# The Piuquencillo Fault System: a long-lived, Andean-transverse fault system and its relationship with magmatic and hydrothermal activity

José Piquer[1], Orlando Rivera[2], Gonzalo Yáñez[3,4], Nicolás Oyarzún[1]

[1]Instituto de Ciencias de la Tierra, Universidad Austral de Chile, Valdivia, 5090000, Chile
[2]Minera Peñoles de Chile, Santiago, 8320000, Chile
[3]Departamento de Ingeniería Estructural y Geotécnica, Pontificia Universidad Católica de Chile, Santiago, 8320000, Chile
[4]Núcleo Milenio Trazadores de Metales (NMTM), Santiago, 8320000, Chile

*Correspondence to*: José Piquer (jose.piquer@uach.cl)

**Abstract.** Lithospheric-scale fault systems control the large-scale permeability in the Earth's crust and lithospheric mantle,
and its proper recognition is fundamental to understand the geometry and distribution of mineral deposits, volcanic and plutonic complexes and geothermal systems. However, their manifestations at the current surface can be very subtle, as in many cases they are oriented oblique to the current continental margin and to the axis of the magmatic arc, can be partially obliterated by younger, arc-parallel faults, and can also be covered by volcanic and sedimentary deposits, through which the fault might propagate vertically.

The Piuquencillo Fault System (PFS) is a proposed lithospheric-scale fault system, located in the Main Cordillera of Central Chile. Here we present the results of the first detailed field study of the PFS, based on structural data collected at 82 structural stations distributed across all the Western Main Cordillera. The first published U-Pb zircon ages for the La Obra batholith, which is bounded to the south by the PFS but it is also affected by younger reactivations of it, were obtained. They yielded $20.79 \pm 0.13$ Ma (granodiorite) and $20.69 \pm 0.07$ (monzogranite). Statistical analysis of fault plane data shows that the presence
of the PFS is reflected on a strong preferred NW to WNW strike, with variable dip directions, evident from the analysis of the total fault plane population and also from individual segments of the PFS. In some segments, the presence of major NE to ENE-striking faults which intersect the PFS is also reflected in the preferred orientation of fault planes. Preferred orientations of hydrothermal veins, breccias and dikes show that both the PFS and some ENE-striking faults were capable of channelling hydrothermal fluids and magma. Kinematic and dynamic analysis of fault-plane data reveals that most of the PFS was
reactivated with sinistral $\pm$ reverse kinematics during the Neogene, under a strike-slip to transpressive regime with E- to ENE-trending shortening direction ($\sigma_1$). Detailed kinematic and dynamic analyses were completed for various segments of the PFS and also for the different rock units affected by it.

This study supports the concept that the PFS is a lithospheric-scale fault system, which strongly controlled deformation and the flow of magmas and hydrothermal fluids during the Neogene. The PFS forms part of a larger, margin-transverse structure,
the Maipo Deformation Zone, a continental-scale discontinuity which cut across the entire Chilean continental margin, and which has been active at least since the Jurassic.

## 1 Introduction

Large-scale permeability of the crust and lithospheric mantle, is controlled by the presence of lithospheric-scale fault systems (McCuaig and Hronsky, 2014). This implies that their characterization is fundamental to understand the distribution and geometry of magmatic-hydrothermal systems and the mineral deposits, volcanic complexes and geothermal systems with which they might be related. However, the surface expression of these type of structures is often very subtle. As they correspond to ancient features, in many cases they are oriented oblique to the axis of younger magmatic arcs and can be obliterated by younger, arc-parallel faults. They might also be covered by volcanic and sedimentary deposits through which the fault has to propagate vertically (Fig. 1). These difficulties can be overcome with a multidisciplinary approach, combining detailed, field-based structural mapping with geologic observations at different scales and various types of geophysical datasets; such an approach has led to the recognition of this type of lithospheric-scale structures in different geological settings, improving our understanding about their complex reactivation histories and their relationship with magmatic-hydrothermal activity (e.g., Chernicoff et al., 2002; Gow and Walshe, 2005; Cembrano and Lara, 2009; Acocella et al., 2011; Lanza et al., 2013; Richards et al., 2013; Fox et al., 2015; Febbo et al., 2019).

In the Chilean and Argentinean Andes, several authors have proposed the existence of pre-Andean, lithospheric-scale structures which are oblique to the N-trending present-day continental margin (Salfity, 1985; Rivera and Cembrano, 2000; Chernicoff et al., 2002). It has been suggested that this pre-Andean structures control the distribution of fossil and active magmatic-hydrothermal centres, including those related to porphyry deposits in northern and central Chile (Rivera and Cembrano, 2000; Richards et al., 2001; Piquer et al., 2016; Yáñez and Rivera, 2019) and volcanic and geothermal systems in the southern part of the country (Cembrano and Lara, 2009). Morphologic and seismic data evidence the offshore continuation of these fault systems (Hicks and Rietbrock, 2015). This type of structures also bound uplifted basement blocks and acted as basin-bounding faults during various episodes of extensional deformation during Andean evolution (Yáñez and Rivera, 2019). However, for most of these structures their field expression remains unclear, as they have been inferred from different types of lineaments or alignments, from geophysical data sets and from abrupt regional-scale discontinuities of N-trending geological units. Here we present the first field-based study of the Piuquencillo Fault System (PFS), a proposed lithospheric-scale structure present in the Andes of central Chile, including a characterization of its kinematics and related paleo-stress tensors, for each individual segment of the PFS. Then we complement our results with different types of geological and geophysical data, including U-Pb zircon dating of plutonic units bounded by the PFS, to provide an integral characterization of this fault system. We also discuss the relevance of this long-lived structure during Andean evolution up to this day, including its role as a pathway for magmas and hydrothermal fluids, and its potential seismic hazard for Chile's most densely populated area.

## 2 Geological background

The Andes of central Chile can be subdivided into two parallel, N-trending ranges, the Coastal and Main Cordilleras (Fig. 2). The Main Cordillera was the position of the Paleogene-Neogene magmatic arc, related to active subduction of the oceanic

Farallon/Nazca plate below the western margin of the South American continent (Pardo Casas and Molnar, 1987; Charrier et al., 2002). Consequently, the Main Cordillera at this latitude is composed mostly of Cenozoic volcanic and intrusive rocks, with subordinate sedimentary intercalations. The evolution of this magmatic arc is characterized by the opening and subsequent inversion of the Abanico Basin (Charrier et al., 2002), an intra-arc volcano-tectonic basin formed between the late Eocene and the early Miocene and inverted in specific, short-lived pulses of contractional deformation since the early Miocene (Charrier et al., 2002; Piquer et al., 2017). The widespread volcanic and sedimentary deposits accumulated during the extensional stages of the basin have been grouped into two main stratigraphic units: the Abanico Formation (Aguirre, 1960; Thomas, 1953), in the area to the east of the city of Santiago (Fig. 2), and the Coya-Machalí Formation (Klohn, 1960) in the mountain regions to the south of it (Fig. 2). The thickness of these units is highly variable, although it can reach up to 5 km (Piquer et al., 2017). Volcanic and sedimentary deposits were accumulated in extensional basins, bounded by normal faults. Main basin-bounding faults are N-striking; they dip to the W in the eastern basin margin (El Fierro-Las Leñas-El Diablo faults, Charrier et al., 2002; Farías et al., 2010) and to the E in the western margin (Infiernillo and Pocuro-San Ramón faults, Farías et al., 2010). Individual segments of the Abanico Basin, in turn, are bounded by arc-oblique, NW- and NE-striking fault systems, also active as predominantly normal faults during this period (Piquer et al., 2015, 2017).

Tectonic inversion since the early Miocene was associated with a decrease in the rates of volcanic output and with the emplacement in the upper crust of a series of Miocene to early Pliocene plutons (Fig. 2). Volcanic rocks accumulated during tectonic inversion are grouped into the Farellones Formation in the area to the east of Santiago (Klohn, 1960), and the Teniente Volcanic Complex (TVC) towards the south (Godoy, 1993; Kay et al., 2005; Fig. 2). The final stages of Neogene compression, crustal thickening and magmatism in the late Miocene – early Pliocene produced the growth of differentiated, upper-crustal magmatic-hydrothermal complexes, which led to the formation of two giant porphyry Cu-Mo deposits: Río Blanco-Los Bronces and El Teniente (Fig. 2). After the formation of these two mineral deposits, the magmatic arc migrated ~40 km towards the east. During tectonic inversion, the N-striking, basin-bounding faults of the Abanico Basin were reactivated as high-angle reverse faults (Charrier et al., 2002; Giambiagi et al., 2003; Farías et al., 2010), while the arc-oblique fault systems which segmented the basin, were reactivated as strike-slip faults with a variable reverse component (Piquer et al., 2015, 2016).

The NW to WNW-striking PFS was defined during regional-scale geological studies around the El Teniente Cu-Mo porphyry deposit (Rivera and Cembrano, 2000; Rivera and Falcon, 2000). These authors indicated that the fault has an average strike of N60°W, dipping 70-80° towards the south. It was proposed that the fault acted as a basin-bounding fault during extensional deformation related to the opening of the Abanico Basin. The Coya-Machalí Formation, which according to the authors has a larger sedimentary component than the also syn-extensional Abanico Formation, appeared to be present only to the south of the PFS. A set of sheeted dikes was emplaced later along different branches of the PFS. Later studies, based on U-Pb zircon geochronology and whole-rock geochemistry, confirmed that the PFS constitute a major boundary between two contrasting segments of the Abanico Basin (Piquer et al., 2017; Fig. 2). The two segments are characterized by different stratigraphic units and also showing major differences in their tectonic evolution and exhumation history. In the northern segment, which contains the Río Blanco-Los Bronces porphyry, stratigraphic units correspond to the Abanico Formation (~34-22 Ma) and the Farellones

Formation (~22-16 Ma), commonly separated by angular unconformities. In the southern segment, the Abanico Formation is also present, but is covered by the younger Coya-Machalí Formation (23-13 Ma), which is in turn covered by the sub-horizontal volcanic deposits of the TVC (13-6 Ma; Piquer et al., 2017). Rare Earth Elements (REE) patterns in igneous rocks also show major differences between the northern and southern segments (Piquer et al., 2017). This can be observed in the depletion of HREE documented in lava flows of the Farellones Formation, not observed in the coeval deposits of the Coya-Machalí Formation, and also in middle Miocene plutons emplaced in both segments, which also show a clearly stronger depletion of HREE and steeper REE patterns in the northern segment. This suggest that the northern segment was affected by compressive deformation and crustal thickening earlier than the southern segment.

## 3 Methodology

Figure 3 shows the distribution of structural stations within the PFS and surrounding areas. Data was collected during three field campaigns, between 2016 and 2018. From the 82 structural stations shown in Fig. 3, 240 fault planes were measured, 54 of them containing kinematic information. The parameters measured in each fault plane were strike and dip, rake of striation, hydrothermal mineral infill (when present) and, when possible, sense of movement based on different kinematic criteria for brittle faults (Fig. 4). Most kinematic indicators correspond to steps in syn-tectonic hydrothermal mineral fibers, and syn-tectonic minerals precipitated in strain fringes (Fig. 4). When syn-tectonic minerals were absent, sense of movement was established by RM and P-only criteria (Petit, 1987; Fig. 4) and by offset markers. Additionally, the orientation of 50 dikes and 109 veins and hydrothermal breccias were measured. The complete structural database is available as Supplementary material. Preferred orientations of faults, dikes and veins were analyzed using the software Stereonet (Allmendinger et al., 2012). Kinematic and dynamic analyses of the fault plane database were also completed. The aim of the first one is to establish the orientation of the pressure and tension axes for each individual fault plane and the average kinematic axes (shortening, stretching and intermediate axes) for different fault populations. This was achieved using the FaultKin software (Allmendinger et al., 2012). Regarding the dynamic analysis, the Multiple Inverse Method (Yamaji, 2000) was used to calculate the orientation of paleo-stress tensors from the inversion of fault-slip data. The advantage of this method is that it allows the identification of separate stress states from heterogeneous data sets. A stress state is defined by four parameters: the orientation of the three principal stresses ($\sigma_1$, $\sigma_2$, $\sigma_3$) and the stress ratio $\Phi = (\sigma_2 - \sigma_3)/(\sigma_1 - \sigma_3)$. The stress ratio varies from 0 to 1, and describes the shape of the stress ellipsoid.

Two U-Pb LA-ICP-MS zircon ages were obtained at the Geochronology Laboratory of SERNAGEOMIN, the Chilean geological survey. Analytical procedures are detailed in Appendix 1.

## 4 Results

### 4.1 Structural setting

The structural database obtained from the 82 structural stations was used to establish the preferred orientations of fault planes, veins and dikes, and also to complete kinematic and dynamic analyses with the aim of establishing the prevailing strain axes

and stress tensors in different segments of the PFS and other nearby faults, during successive reactivation events.

To achieve this aim, the study area was subdivided into five sectors (Fig. 5), each of them characterized by specific lithotypes and structural patterns.

The Clarillo-La Obra sector (Fig. 5) includes structural stations located along one of the main WNW-striking branches of the PFS; the affected lithologies correspond to the lower Miocene La Obra batholith and the Abanico Formation volcanic rocks.

The La Obra batholith corresponds to a major intrusive complex which is bounded to the south by the PFS but it is also affected by younger reactivations of it. Two main facies of the batholith were recognized. One of them corresponds to an equigranular, medium- to coarse-grained granodiorite, with abundant biotite and minor hornblende. Some biotite crystals show weak chlorite alteration at their margins. The granodiorite contains common rounded enclaves of diorite composition, composed of fine-grained plagioclase, hornblende and magnetite. This is the most typical lithofacies of the intrusive complex. The second major

lithofacies correspond to a hornblende-rich quartz-monzonite, with minor biotite. Both mafic minerals are variably altered to chlorite. Graphic texture of alkali feldspar and quartz is common. This unit is finer grained than the granodiorite, and it is exposed at the westernmost outcrops of the batholith.

The Piuquencillo alto and Piuquencillo-Claro sectors (Fig. 5) are located in the Piuquencillo river valley, where the PFS was defined (Rivera and Falcon, 2000). In the Piuquencillo alto sector, the main lithological unit are different intrusive facies of

the late Miocene Carlota Intrusive Complex (CIC), which is emplaced in volcanic rocks of the middle Miocene TVC. The main facies of the CIC is a hornblende- and biotite-rich granodiorite, with variable degrees of hydrothermal alteration. This facies was dated by Kurtz et al. (1997) at $8.7 \pm 0.3$ Ma ($^{40}$Ar/$^{39}$Ar in biotite). Because of the evidences of relatively high-temperature hydrothermal alteration (secondary minerals such as epidote, chlorite, actinolite and muscovite), it is likely that this age reflects the time of hydrothermal activity, not of magmatic crystallization. Diorites, quartz-diorites, monzodiorite

porphyries and quartz-monzonites were also observed within the CIC. The Piuquencillo-Claro sector, in turn, is dominated by volcanic and volcaniclastic rocks of the Abanico and Coya-Machalí Formations, intruded by small-scale andesitic and daci-andesitic stocks and dikes. According to regional-scale stratigraphic interpretations (Piquer et al., 2017) the PFS correspond to the transition zone between these two units, with the Coya-Machalí Formation present only towards the south of this structure. There are no absolute ages for the dike sets, but because of the age of their host rocks, they are post-early Miocene

in age. No dikes of this type were observed cross-cutting the altered CIC rocks in the Piuquencillo Alto sector, from which we infer they were emplaced before the late Miocene. However, several individual generations of dikes might have been emplaced during this timeframe, and they might have acted as feeder for the Teniente Volcanic Complex.

The San Pedro de Nolasco and Maipo sectors are located along a less notorious, WNW-striking branch of the PFS, and also contain traces of major NE-striking faults (Fig. 5). The San Pedro de Nolasco sector in particular includes the Ag-Pb-Zn-Cu

vein system of the same name. The strike of individual veins varies from ENE to WNW, but the whole vein system defines a WNW-trending belt, 1.5 km long and 200 m wide. The veins are composed of quartz, calcite and barite with a sulphide ore of

galena, sphalerite, tennantite/tetrahedrite, chalcopyrite and bornite (Leal, 2018), and they are syn-tectonic, as demonstrated by several evidences of hydrothermal mineral crystallization during fault slip (Fig. 4). The polymetallic veins of San Pedro de Nolasco are emplaced in volcanic rocks of the TVC (13-6 Ma; Piquer et al., 2017), which in the area overlay volcanic rocks of the Farellones Formation (Fig. 6). From this, a maximum middle Miocene age can be assigned to the vein system. Considering that the youngest evidences of hydrothermal activity in the Neogene magmatic arc of central Chile occur at ~4 Ma (Maksaev et al., 2004; Deckart et al., 2013, 2014), the age of the vein system can be constrained to the middle Miocene – early Pliocene. The Maipo sector, in turn, contains mostly Abanico Formation volcanic rocks, intruded by small-scale stocks and dikes, some of them probably related to the middle Miocene San Gabriel pluton, located towards the north.

### 4.2 U-Pb zircon dating

Two new U-Pb zircon ages were obtained for this study (Table 1, analytical data in Appendix 2), from samples collected from different facies of the La Obra batholith, from which no previous U-Pb crystallization ages have been documented. Tera-Wasserburg plots for the dated samples are shown in Fig. 7. The two samples are representative of the two main facies of the La Obra batholith: sample FP01 comes from the biotite-rich granodiorite, while sample FP03 comes from the hornblende-rich quartz-monzonite. The granodiorite sample yielded an age of $20.79 \pm 0.13$ Ma. The calculated age of the quartz-monzonite, in turn, is $20.69 \pm 0.07$ Ma, almost identical to the main granodiorite body. These ages confirm that La Obra is the southernmost lower Miocene intrusive complex of central Chile; to the south of the PFS, all the outcropping plutonic complexes in the Main Cordillera are middle Miocene or younger (Piquer et al., 2017, and references there in).

### 4.3 Preferred orientations

Figure 8 illustrates the preferred orientations of dikes/main intrusive contacts, veins/main hydrothermal breccia contacts, and faults.

Dikes and intrusive contacts, and also veins and hydrothermal breccias show strikes varying from ENE to NNW, with a remarkable absence of strikes approaching a N-S orientation. Regarding orientation of fault planes, the strong influence of the PFS is clearly visible, with a remarkable WNW preferred orientation, parallel to the general tendency of the PFS. Also visible is a secondary trend of ENE-striking fault planes, and a minor group of NNW-striking faults. Similar to the case of the veins and hydrothermal breccias, there is a remarkable scarcity of N-S striking fault planes.

Figure 9 shows the preferred orientations of fault planes for the five different sectors into which the study area was subdivided for statistical analyses. Although the presence of large populations of WNW-striking fault planes related to the PFS is evidenced in all the five sectors, some remarkable differences are visible between them. The Piuquencillo-Claro sector, where the PFS was defined, shows the strongest dominance of WNW-striking fault planes. The Piuquencillo Alto and San Pedro de Nolasco sectors clearly show the influence of an ENE-striking fault system, apart from the PFS. The Clarillo-La Obra sector marks the presence of a NNE-striking set of fault planes. Finally, in the Maipo area, the influence of a NE-striking fault system is evident, which probably correspond to the Yeso Valley Fault System (Fig. 2), a major NE-striking, dextral strike-slip fault.

In 54 fault planes it was possible to obtain reliable kinematic information. A variety of syn-tectonic hydrothermal minerals were observed, including tourmaline, calcite, hematite, epidote and actinolite (Fig. 4). They are particularly common within and in the vicinity of plutonic complexes. The orientation of all the 54 fault planes, their slickenlines and sense of movement are shown in Figure 10, while Figure 11 shows the same information for each individual sector. Clarillo-La Obra sector is not shown, as no reliable kinematic data was obtained in this area. Preferred orientations of fault planes with kinematic information

(Figs. 10, 11) are similar to the ones obtained from the total fault plane database (Figs. 8, 9), with a strong WNW preferred orientation, a secondary group striking ENE, and a minor population of fault planes striking NNW. Slickenlines most commonly show low pitch values (Fig. 10), indicating predominantly strike-slip movements.

## 5 Discussion

### 5.1 Kinematic and dynamic analysis

When considering all the 54 fault planes for which there is kinematic information available, it is evident that the sense of movement of most of the faults is consistent with fault activity under a strike-slip regime, with E-W to ENE-directed, sub-horizontal shortening and N-S to NNW-directed, sub-horizontal stretching. This is shown by the orientation of the average pressure and tension axis in the kinematic analysis and by the main clusters of $\sigma_1$ and $\sigma_3$ in the dynamic analysis (Fig. 10). We interpret this as the predominant regional stress state during the middle Miocene to early Pliocene, considering the ages of the

different rock units affected by the faults, and the age range of hydrothermal activity in the Neogene magmatic arc of central Chile, which constrains the age of syn-tectonic mineral fibres used to obtain kinematic information in fault planes. However, the kinematic analysis (Fig. 10) shows an important dispersion of individual pressure and tension axes, suggesting a relatively heterogeneous deformation, in which the movement of several fault planes is not compatible with the average shortening (pressure) and stretching (tension) axes. The dynamic analysis, using the Multiple Inverse Method (Fig. 10), allows the

distinction of secondary clusters of $\sigma_1$ and $\sigma_3$, both of them vertical, showing that some groups of fault planes were active under extensional and compressional conditions respectively.

As discussed before, and shown by Figure 8, most of the measured fault planes have a WNW orientation, while the veins are more evenly distributed between WNW and ENE orientations. This could be related to the predominant ENE trend of $\sigma_1$: as the ENE faults are more parallel to $\sigma_1$, they are more efficient as fluid pathways, in contrast to WNW to NNW-striking faults,

which are at higher angles relative to $\sigma_1$ and require higher fluid pressure or transient stress relaxation to open and allow the circulation of fluids.

Similar kinematic and dynamic analyses were completed for the five sectors shown in Figure 5, and also for different lithological units, to explore temporal variations in the stress state and strain axis, by looking at the variability in the results of the analysis for faults cross-cutting rocks of different ages. The lithological units considered were the Abanico and Coya-

Machalí Formations (upper Eocene – middle Miocene); the Farellones Formation (lower to middle Miocene) and the Teniente

Volcanic Complex (middle to upper Miocene); subvolcanic intrusions (middle to upper Miocene); and the Miocene plutons. The results of this analyses are presented in Figures 12 to 15.

The kinematic and dynamic analysis of fault plane data by sector shows some remarkable differences between the upper and lower part of the Piuquencillo river valley. In the upper part of the valley (Piuquencillo alto), faults were active under a pure strike-slip regime (Fig. 13), while in the lower part (Piuquencillo-Claro), a transpressive (transitional between strike-slip and compressive) tectonic regime was predominant, showing very low Φ values and a large variability in the orientation of σ3 (Fig. 13). This could be reflecting a trend from purely strike-slip regime in the central part of the Abanico/Coya-Machalí basin, to a transpressive regime predominant closer to the basin margins, a feature already observed in regional studies (Piquer et al., 2016) and which might be due to an excess of gravitational potential energy in the central part of the Main Cordillera, at the axis of the Paleogene-Neogene magmatic arc.

When interpreting the results of stress tensor calculations from the inversion of fault slip data, a possibility which has to be considered is that stress tensor rotations might occur in the vicinity of major faults, although the expected patterns of stress rotation are still a matter of debate (Hardebeck and Michael, 2004; Famin et al., 2014). If these stress tensor rotations occurred at the PFS, then part of our calculations might not represent a regional stress field, but a local stress tensor acting only in and around the fault traces. The dynamic analysis by sector, however, suggests that this is not the case. The Piuquencillo Alto and Piuquencillo-Claro sectors cover areas around the main traces of the PFS, while the Maipo sector is located further away from it (Fig. 13). In these three sectors, the direction of maximum horizontal compression is similar (E-W to slightly ENE), without any evidence of rotations occurring around the traces of the PFS. The results obtained are also consistent with regional calculations of the Miocene – early Pliocene stress field in central Chile (Piquer et al., 2016). All of this is consistent with the fact that none of the cropping-out branches of the PFS is individually a major fault; the strain associated to each of them is of small magnitude, so no major perturbations of the stress tensor are expected around them.

When considering the kinematic and dynamic analysis by lithological units, it is observed that the largest variability in the orientation of the principal stresses is observed in the volcanic units (Fig. 14), probably reflecting local variations in the stress state. The tectonic (far-field) stress tensor is more clearly defined in the intrusive units. The differences in the Φ value (Fig. 14) directly reflect the geographic position of the intrusive units: the predominant stress tensor calculated for faults in the Miocene plutons is identical to the one calculated for the Piuquencillo alto sector (Fig. 13), while the stress tensor calculated for subvolcanic intrusions is very similar to the one calculated for the Piuquencillo-Claro sector, where most of those intrusions are located.

## 5.2 Fault orientations and the flow of magmas and hydrothermal fluids

As discussed before and shown by Figure 8, it is evident that both WNW-striking faults, belonging to the PFS, and ENE-striking faults were capable of channelling magmas and hydrothermal fluids, as reflected in the preferred orientations of dikes and, particularly, hydrothermal veins and breccias. Syn-mineral displacement of the faults was mainly dextral for ENE- to NE-striking faults and sinistral for WNW-striking faults (Fig. 15). This suggests that the PFS and the ENE-striking faults acted

broadly as conjugate faults under the prevailing middle Miocene – early Pliocene stress tensor. However, they are not oriented at the ideal angle with respect to $\sigma_1$ expected in intact rocks, with the ENE-striking faults being more parallel to $\sigma_1$ than the fault planes of the PFS. The reason for this might be that both sets of faults are part of large-scale, pre-existing fault systems (the PFS and, for the ENE-striking faults, the Yeso Valley Fault System), reactivated during the Mio-Pliocene, but not originated as conjugate structures. As the ENE-striking faults are more parallel to the predominant orientation of $\sigma_1$ (E-W to ENE-trending), they were the most favourably oriented for opening, which explains why ENE-striking veins are as common as those striking WNW, while ENE-striking fault planes are much less frequent (Fig. 8). NNW-striking fault segments of the PFS in particular are the least favourable for opening under the predominant stress regime. This could make them highly attractive for mineral exploration, as they will tend to remain sealed for large periods of time, accumulating volatiles and allowing magmas to differentiate at depth, until fault reactivation occurs (perhaps triggered by co-seismic stress relaxation leading to transient local extension, as suggested by Mpodozis and Cornejo, 2012) creating instant permeability along the fault and allowing the rapid (often catastrophic) ascent of differentiated magmas and hydrothermal fluids. This is very clearly observed in the Río Blanco-Los Bronces porphyry Cu-Mo cluster, in which hydrothermal breccias and dacitic porphyries are emplaced along a NNW-striking fault system, while most of the late quartz veins and andesitic dikes are emplaced along NE-striking faults, more favourable for opening (Mpodozis and Cornejo, 2012; Piquer et al., 2015).

**5.3 Beyond the PFS: the Maipo Deformation Zone**

Our new field data demonstrate that the PFS can be traced across the entire Western Main Cordillera (Fig. 16) of Central Chile, confirming the proposition of Rivera and Cembrano (2000). However, Yáñez et al. (2002) proposed that the PFS might be part of a larger, continental-scale discontinuity, the Maipo Deformation Zone (MDZ; Fig. 16). Several recent works completed in the coastal ranges near Valparaíso (Fig. 16), after the work of Yáñez et al. (2002) was published, confirm that the PFS can be extended to the NW across the entire continental margin. Evidences for the existence of a deep, long-lived, NW-striking fault system in the coastal ranges are varied. Creixell et al. (2011) showed that different Jurassic intrusions were syn-tectonically emplaced along NW-striking faults, under both sinistral transtension and transpression, while Hernández (2006) documented primitive, Mesozoic mafic and ultramafic rocks of mantle origin emplaced along similar faults. In the same area, the structural architecture of the Upper Jurassic Antena Au vein district (Townley et al., 2000), is also dominated by NW-striking faults, which control the location of the mineralized district together with a system of conjugate, NE-striking faults. More to the SE, a similar situation occurs at the Early Cretaceous Lo Aguirre stratabound Cu deposit (Fig. 16): the orebody has a very strong NW elongation, while post-mineral faults strike NE (Saric et al., 2003). The latter appear to be also part of a large-scale structure, as they are on strike of a set of major NE-striking faults identified in the Western Main Cordillera (the Saladillo, Flores and El Salto fault systems, Piquer et al., 2015), in the vicinity of the Río Blanco-Los Bronces porphyry Cu-Mo cluster (Fig. 16). Subsequently, Rivera (2017) showed that this NW-striking fault system is associated with regional-scale geological discontinuities in the coastal ranges. To the north, there is a continuous N-trending belt of Jurassic sedimentary and volcanic units (Ajial, Cerro Calera and Horqueta Formations), and Paleozoic rocks are absent. To the south, outcrops of Jurassic

stratigraphic units and plutons are highly discontinuous, and the geology of the coastal ranges is dominated by upper Paleozoic intrusions and isolated blocks of metamorphic rocks (SERNAGEOMIN, 2002; Rivera, 2017; see Fig. 16). Yáñez and Rivera (2019) proposed the existence of a series of continental-scale discontinuities in the Chilean continental margin, which they

defined as TLFs (Trans-Lithospheric Faults). The concept of a TLF is equivalent to the fundamental basement structures of McCuaig and Hronsky (2014). Even though the whole lithosphere involvement of these continental-scale structures has not been demonstrated empirically in the Andes, several lines of reasoning provide independent arguments to support this concept. In one hand, Yáñez and Rivera (2019) proposed that the origin of these deep-seated structures could be related with master and transform faults associated with ancient rifting and/or suture zones related to collisional processes. In both likely scenarios,

recent analogues show the presence of deep seated structures that involve the whole lithosphere (i.e. Kuna et al., 2019; Hua et al., 2019). On the other hand, several authors have demonstrated that continental-scale deformation zones, of some hundreds of kilometres length (comparable in scale to the PFS/MDZ), are controlled by the rheology of the mantle (i.e. Bird and Piper, 1980, England and McKenzie, 1984). Moreover, numerical models, in agreement with field observations, indicate that deformation decays laterally to 1/10 of the structure length for strike-slip dominated movements (England et al., 1985), thus

developing a deformation zone of 10-30 km width, similar to the PFS/MDZ. According to the interpretation of Yáñez and Rivera (2019), Lo Aguirre was emplaced at the intersection of two TLFs, at the margins of a dense crustal block observed in regional gravity data: the Valparaíso-Volcán Maipo TLF, which coincides with the Maipo Deformation Zone (including the PFS), and the Aconcagua-San Antonio TLF, which includes the NE-striking fault systems identified in Lo Aguirre and at the Río Blanco-Los Bronces district. These areas of intersection of major fundamental basement structures develop complex

interference patterns dominated by abundant secondary faults and fractures, and therefore are associated with high permeability, being favorable sites for the emplacement of mineral deposits. Also, it is common that these intersecting, continental-scale faults define wedge-shaped blocks with distinctive stratigraphy and internal deformation styles (Piquer et al., 2019).

Within the study area, at the Clarillo-La Obra and Piuquencillo/Piuquencillo-Claro sectors (Fig. 5), the PFS coincides with a

major change in the deformation style of the Cenozoic infill of the inverted Abanico Basin. Towards the south, the Miocene Coya-Machalí Formation crop out, and this unit is strongly deformed by a series of tight folds and reverse faults, constituting the Cordón Perales fold and thrust belt (Rivera, 2017). The fold axes of these folds are truncated by the PFS, and to the north of it, the Coya-Machalí Fm. is absent (Fig. 2) and the older, Eocene-Oligocene Abanico Formation is more gently folded.

Towards the SE of the study area, the PFS is associated with major changes in the orientation and vergence of the N- to NNE-

striking fault systems that define the boundary between the Western and Eastern Main Cordillera, at the inverted eastern margin of the Paleogene Abanico Basin (Fig. 16; Rivera, 2017). Moreover, the seismic activity related to these arc-parallel faults is much more intense to the south of their intersection with the PFS than to the north of it, and a very large cluster of seismic activity appears at the intersection zone (see Fig. 3 of Piquer et al., 2019). Nevertheless, we cannot rule out other reasons to explain the differences in seismic activity, among them differences in recording time window and/or differences in water

percolation between both sectors. Further SE, in the Eastern Main Cordillera, the prolongation of the PFS is at least spatially

related to the Escalones prospect (Fig. 16), a Cu skarn deposit emplaced in Lower Cretaceous marine sedimentary rocks, with a well-developed skarn alteration mineralogy, and high hypogene Cu grades (Maksaev et al., 2007). There is no published information about the local-scale structural controls on mineralization at Escalones, but the Lower Cretaceous calcareous beds strike NW, parallel to the PFS (Maksaev et al., 2007). To the SE of Escalones, the PFS follows the northern margin of the Diamante Caldera, a major Pleistocene collapse structure (Stern et al., 1984; Harrington, 1989) within which the Maipo stratovolcano (Fig. 16) is located.

There is also evidence that some segments of the MDZ are tectonically active. Sabaj (2008) identified and characterized potentially active faults in the Coastal ranges of central Chile. It was concluded that fault architecture in the area is dominated by NW-striking faults, intersected by different sets of NE-striking faults. At least four of the individual NW-striking faults recognized in the work of Sabaj (2008) are located within the Maipo Deformation Zone, on-strike of the PFS: the Marga-Marga, Valparaíso, Laguna Verde and Valparaíso-Curacaví faults. They were later grouped in the Valparaíso Fault System (VFS) by Del Valle (2018). Both the NW- and NE-striking faults were concluded to be potentially active by Sabaj (2008), although the NW-striking faults were considered to pose the higher risk, as their traces are more continuous and longer. It was estimated that the maximum possible magnitude (Mw) of seismic events generated by the NW-striking faults is between 5.8 and 7.1. The seismic hazard posed by these NW-striking faults in the Andean forearc was confirmed by the activation of the Pichilemu fault, a regional-scale fault located ~140 km to the south of the MDZ, after the Mw 8.8 subduction earthquake of 27 February 2010 (Farías et al., 2011; Aron et al., 2013). The Pichilemu fault was activated on 11 March, 12 days after the main interplate earthquake, and produced two main shocks, Mw 6.9 and 7.0 (Farías et al., 2011; Aron et al., 2013). Focal mechanisms indicate normal movement of the SW-dipping Pichilemu fault (Farías et al., 2011; Aron et al., 2013), consistent with the expected relaxation of NW-striking crustal faults (normally under compression) during the co- and post-seismic periods, as was also observed in the volcanic arc at the Main Cordillera (Spagnotto et al., 2015). In the Main Cordillera, there is no direct evidence of neotectonic activity of the PFS, but similar, WNW-striking faults have been shown to displace Quaternary terraces of the Maipo river, ~25 km to the north of the PFS (Lavenu and Cembrano, 2008). One of the focal mechanism solutions of the Las Melosas earthquake, a major (Mw 6.9) intraplate seismic event registered in 1958 in the Main Cordillera, less than 10 km to the north of the PFS, is compatible with activity along a WNW-striking fault (Alvarado et al., 2009). If this was the case, it is most likely that this earthquake was generated by one of the northernmost branches of the PFS, although this particular fault reactivation would involve a dextral strike-slip sense of movement (Alvarado et al., 2009), not sinistral as was the predominant sense of movement during the Mio-Pliocene (Fig. 15). However, the other possible solution, represented by a NNE-striking fault plane, is equally plausible, considering the presence of several individual faults with NE- to NNE-strike near Las Melosas (Piquer et al., 2019).

Considering the fact that both the PFS in the Main Cordillera and the VFS in the fore-arc are potentially active, it is worth noting that the southern part of the city of Santiago is built on top of the MDZ, between the PFS and the VFS. The Santiago valley is covered by unconsolidated sedimentary and volcanic deposits (SERNAGEOMIN, 2002; Fig. 16). Yáñez et al. (2015), based on detailed gravimetric modelling, showed that the topography of the basement of the Santiago valley presents large-

scale breaks and scarps which coincide with the expected position of major branches of the MDZ, and also with other, NE-striking faults. The assessment of the neotectonic activity of the MDZ-related faults, their recurrence intervals and the risk they pose to the Santiago and Valparaíso urban areas are highly relevant topics, which require further studies and evaluation. As shown in Figure 16, there is a second continental-scale structural system sub-parallel to the PFS and the Maipo Deformation Zone, located immediately southward. This second fault system, from east to west, defines the southern boundary of the

Diamante caldera, it has been well recognized at the El Teniente porphyry Cu-Mo district (Piquer et al., 2016; Fig. 16), and it is also well defined at the Coastal ranges and plains of central Chile (SERNAGEOMIN, 2002; Fig. 16), passing through the port city of San Antonio. Here we will refer to this structure as the Teniente-San Antonio fault system. This major fault system and the Maipo Deformation Zone might be the manifestations in the present-day surface of the same continental scale discontinuity at depth; however, testing this possibility is beyond the scope of this work.

There are also geophysical evidences that highlights the relevance of the PFS, the broader Maipo Deformation Zone, and also the Teniente-San Antonio fault system. Yáñez et al. (1998) defined the Melipilla anomaly, a large-scale, WNW-striking negative magnetic anomaly located in the coastal area of central Chile (Fig. 17), which coincides with the western segment of the Teniente-San Antonio fault system. To the north of the Melipilla anomaly, a second, NW-striking negative magnetic anomaly is also evident (Fig. 17), and coincides with the western segment of the Maipo Deformation Zone. These negative

magnetic anomalies most likely correspond to felsic Paleozoic and Mesozoic intrusive bodies, emplaced along these long-lived fault systems.

Finally, a different type of manifestation of these continental-scale structures might have been observed during the Valparaíso seismic sequence of 2017 (Nealy et al., 2017). The distribution of the hypocenters of the multiple earthquakes registered during this event (Fig. 17) shows a clear NW alignment, and is neatly bounded by the submarine prolongation of the Maipo

Deformation Zone and the Teniente-San Antonio fault system. The earthquakes are strongly concentrated at the plate boundary mega-thrust (Nealy et al., 2017), so they were not generated by the activation of crustal faults. However, the remarkable spatial relationship between earthquake distribution and large-scale faults in the continental lithosphere shown in Fig. 17, might indicate the existence of a still poorly understood feedback mechanism between large-scale, trans-lithospheric discontinuities in the upper plate and the distribution of subduction-related earthquake sequences at the plate boundary.

**6 Conclusion**

-     The continuity and surface expression of the PFS across all the Western Main Cordillera is confirmed

-     During the middle to late Miocene, the PFS was active with sinistral to sinistral-reverse kinematics, under a strike-slip to transpressive tectonic regime with sub-horizontal, E-W to ENE-trending $\sigma_1$

-     Both the PFS and conjugate, ENE-striking faults, channelled the flow of magmas and syn-tectonic hydrothermal

fluids. The ENE-striking faults were more efficient pathways for hydrothermal fluids than the PFS

-     An early Miocene crystallization age (20.9-20.6 Ma, U-Pb in zircons) is confirmed for the La Obra pluton, which is bounded to the south by the PFS

- The PFS corresponds to the expression, in the Western Main Cordillera, of a continental-scale fundamental basement structure (Fig. 16), which has been called the Maipo Deformation Zone or the Valparaíso-Volcán Maipo trans-lithospheric
fault. This structure has been active at least since the Jurassic, controlling the flow of magmas and hydrothermal fluids, and at least some of its segments are still tectonically active today

- The surface expression of pre-Andean, lithospheric-scale fault systems is often very subtle, but they can be characterized through multidisciplinary studies involving, among others, detailed structural field work and geophysical interpretations. The study of this type of long-lived fault systems is a relevant task, as they strongly control the distribution
and geometry of both fossil and active magmatic-hydrothermal systems, and they can be reactivated with different kinematics during the seismic cycle; the seismic hazard associated with this structures requires more detailed evaluations

**Author contribution**

José Piquer and Nicolás Oyarzún completed the field work and the analysis of fault plane data. Orlando Rivera participated actively in the interpretation of structural and stratigraphic data, providing inputs based in his earlier works at the Piuquencillo
Fault System. Gonzalo Yáñez completed the geophysical interpretations. José Piquer prepared the manuscript with important contributions from the co-authors, mainly Orlando Rivera and Gonzalo Yáñez.

**Acknowledgments**

Founding for the first two field campaigns completed as part of this work came from the DID project S-2016-32, an internal research project from Universidad Austral de Chile titled "El Sistema de Falla Piuquencillo: evolución y control sobre el
emplazamiento de sistemas hidrotermales". The logistics and costs involved on the third field campaign were covered by the Rio Tinto mining company. All the analytical costs were covered by DID project S-2016-32. We sincerely thank Dr. Laura Giambiagi and Dr. Gianluca Vignaroli for their constructive comments and suggestions, which greatly improved the quality of this manuscript.

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

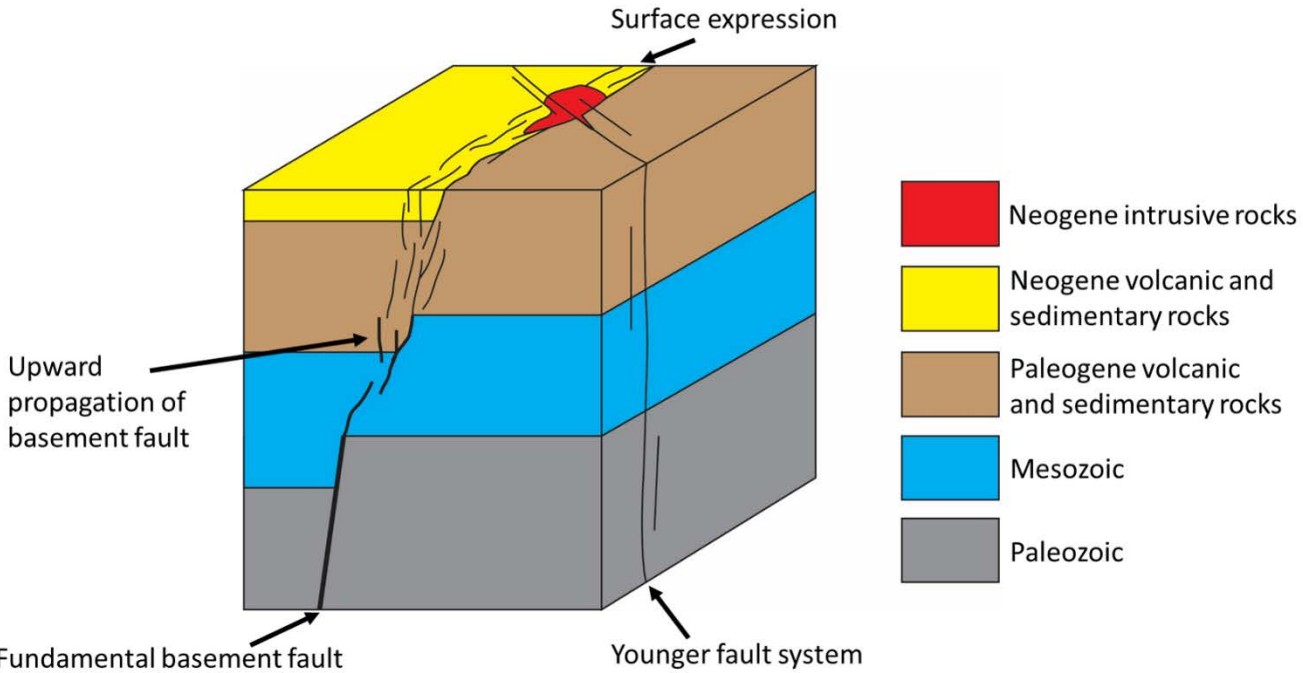

**Figure 1: Schematic diagram showing how Fundamental Basement Structures (FBSs) propagate upwards through younger rocks, being represented at the current surface by a network of minor faults, often difficult to recognize in the field. The diagram also illustrates the common re-activation of this type of faults as basin-bounding faults under extensional conditions, and the relationship between fundamental basement faults and the emplacement of intrusive complexes. A younger, cross-cutting fault system is also illustrated. Adapted from Figure 7 of McCuaig and Hronsky (2014) and Figure 9 of Piquer et al. (2019).**


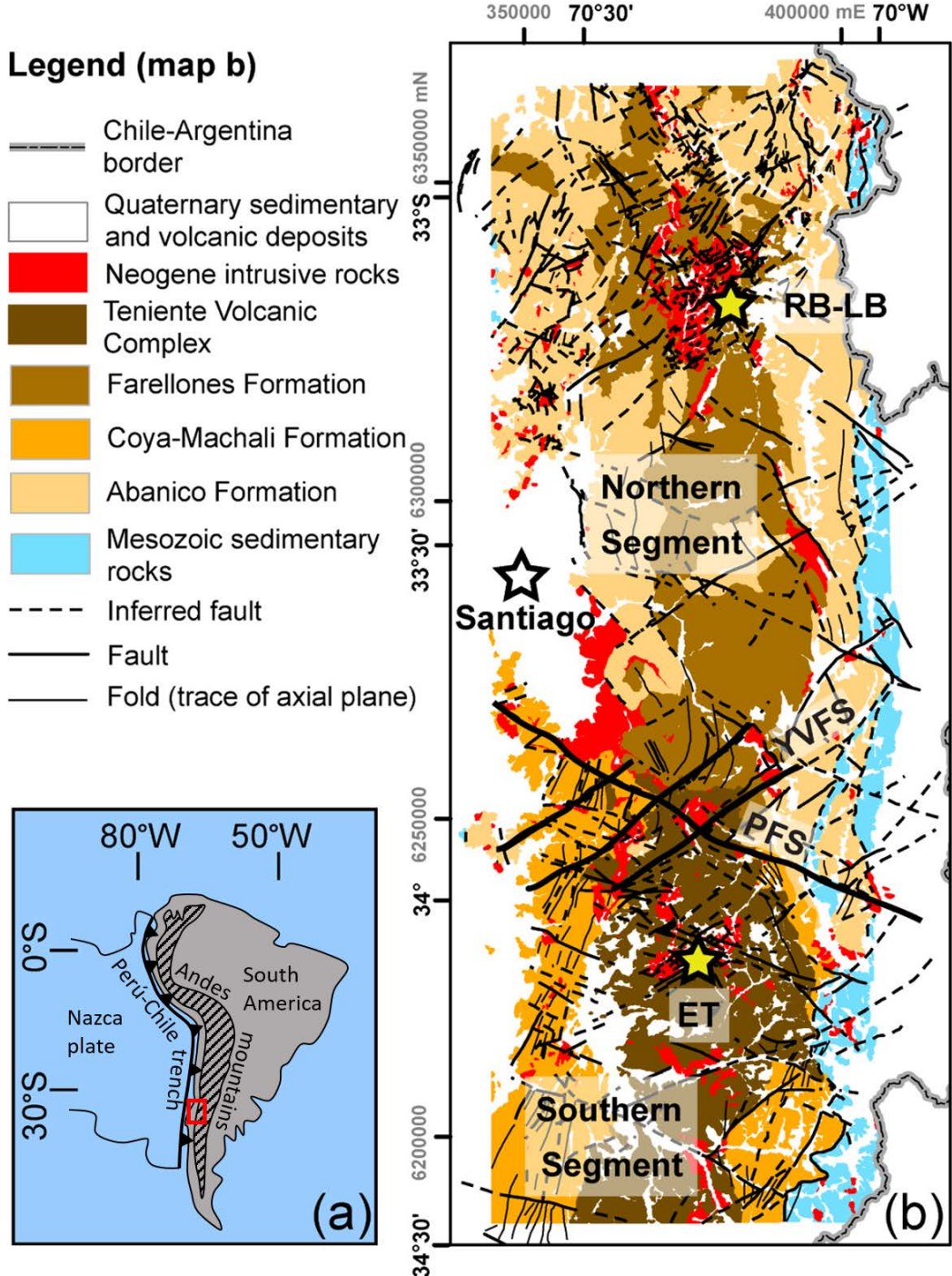

**Legend (map b)**

- — — — Chile-Argentina border
- Quaternary sedimentary and volcanic deposits
- Neogene intrusive rocks
- Teniente Volcanic Complex
- Farellones Formation
- Coya-Machali Formation
- Abanico Formation
- Mesozoic sedimentary rocks
- - - - - Inferred fault
- ——— Fault
- ——— Fold (trace of axial plane)

Figure 2: A. Location of the study area (red rectangle) in South America. B. Geology of the Andes of central Chile, based on Rivera and Falcón (2000), SERNAGEOMIN (2002), Fuentes et al. (2004), Fock (2005) and this work. Quaternary sediments and volcanic deposits not shown. PFS = Piuquencillo Fault System, YVFS = Yeso Valley Fault System, RB-LB = Río Blanco – Los Bronces, ET = El Teniente.

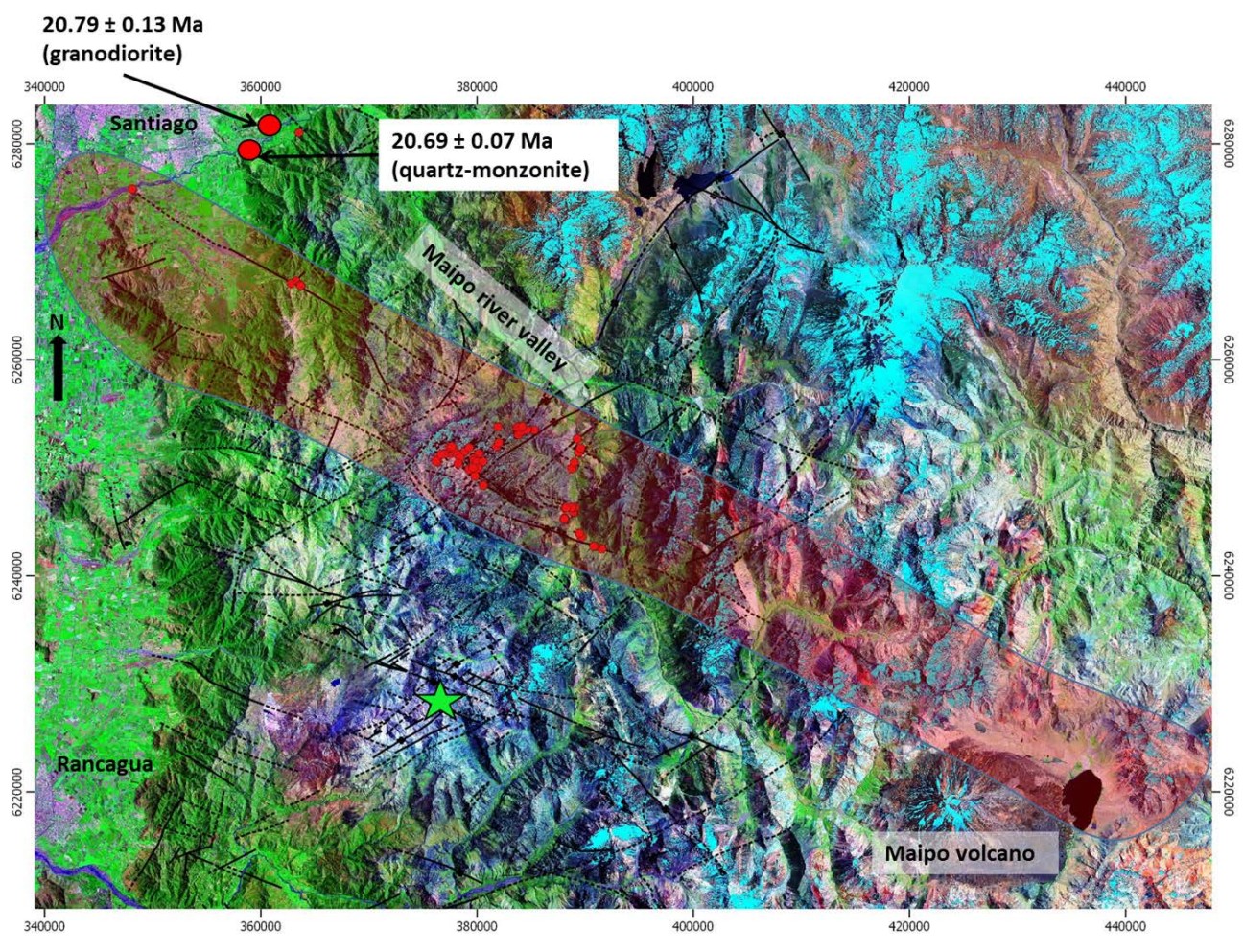


**Figure 3: Distribution of structural stations (red dots) and sample location for U-Pb zircon ages. Semi-transparent red polygon shows the approximate extent of the PFS; green star shows the location of the El Teniente porphyry Cu-Mo deposit. Black lines represent fault traces (continuous = observed, dashed = inferred). The background correspond to a Landsat image, courtesy of the U.S. Geological Survey.**

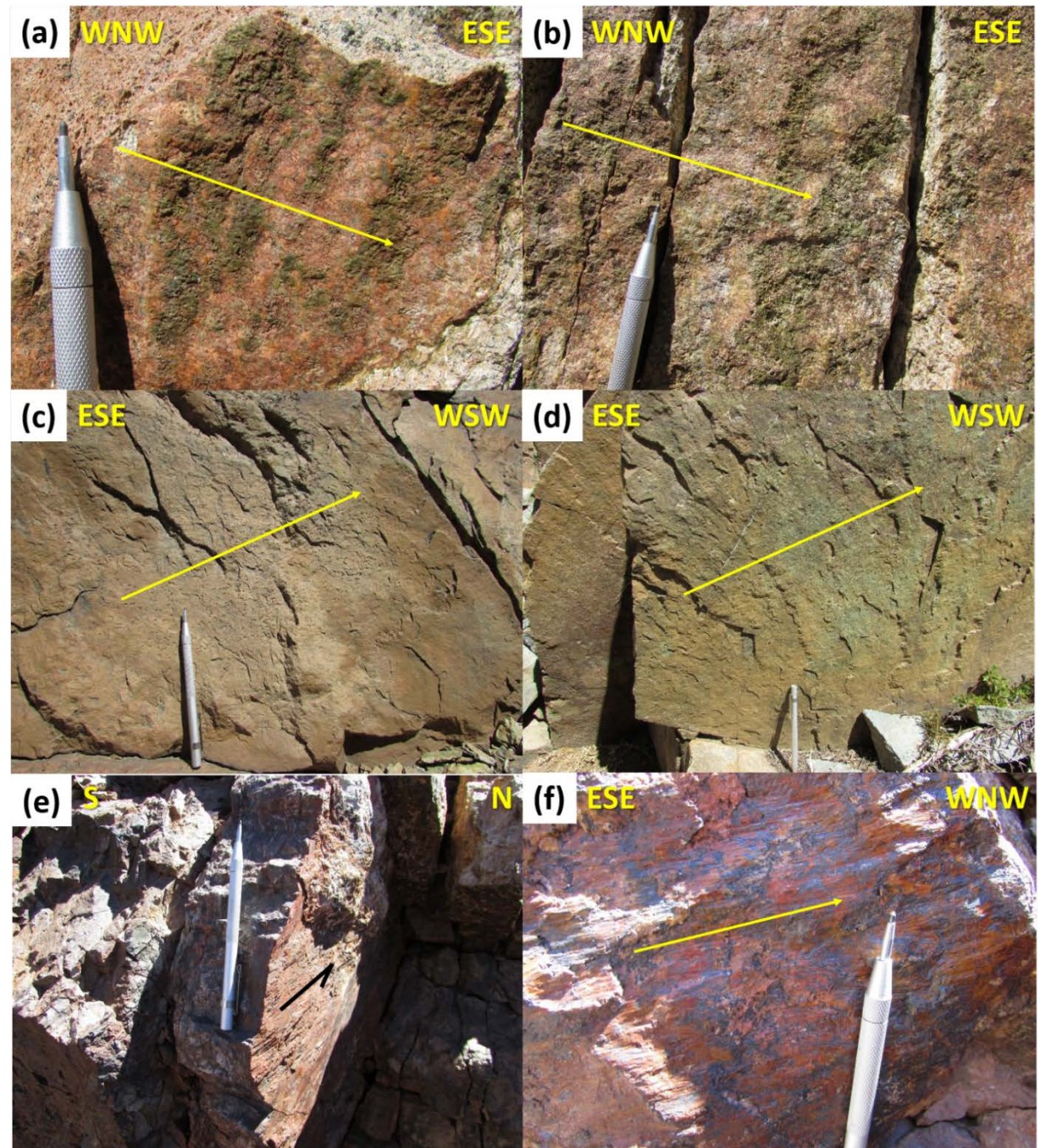

**Figure 4: Fault planes in different branches of the PFS. All of them show evidences of sinistral strike-slip movements. Arrows indicate the sense of movement of the missing block. (a) and (b) correspond to fault planes affecting the Miocene plutons of the Carlota Intrusive Complex, with syn-tectonic epidote (a) and actinolite (b) forming steps and crystallizing in strain fringes. (c) and (d) correspond to fault planes affecting volcanic rocks of the Teniente Volcanic Complex, with sense of movement indicated by RM criteria (Petit, 1987). (e) and (f) illustrate the syn-tectonic, quartz-barite-calcite-hematite San Pedro de Nolasco veins, with the sense of movement indicated by steps in quartz and hematite.**

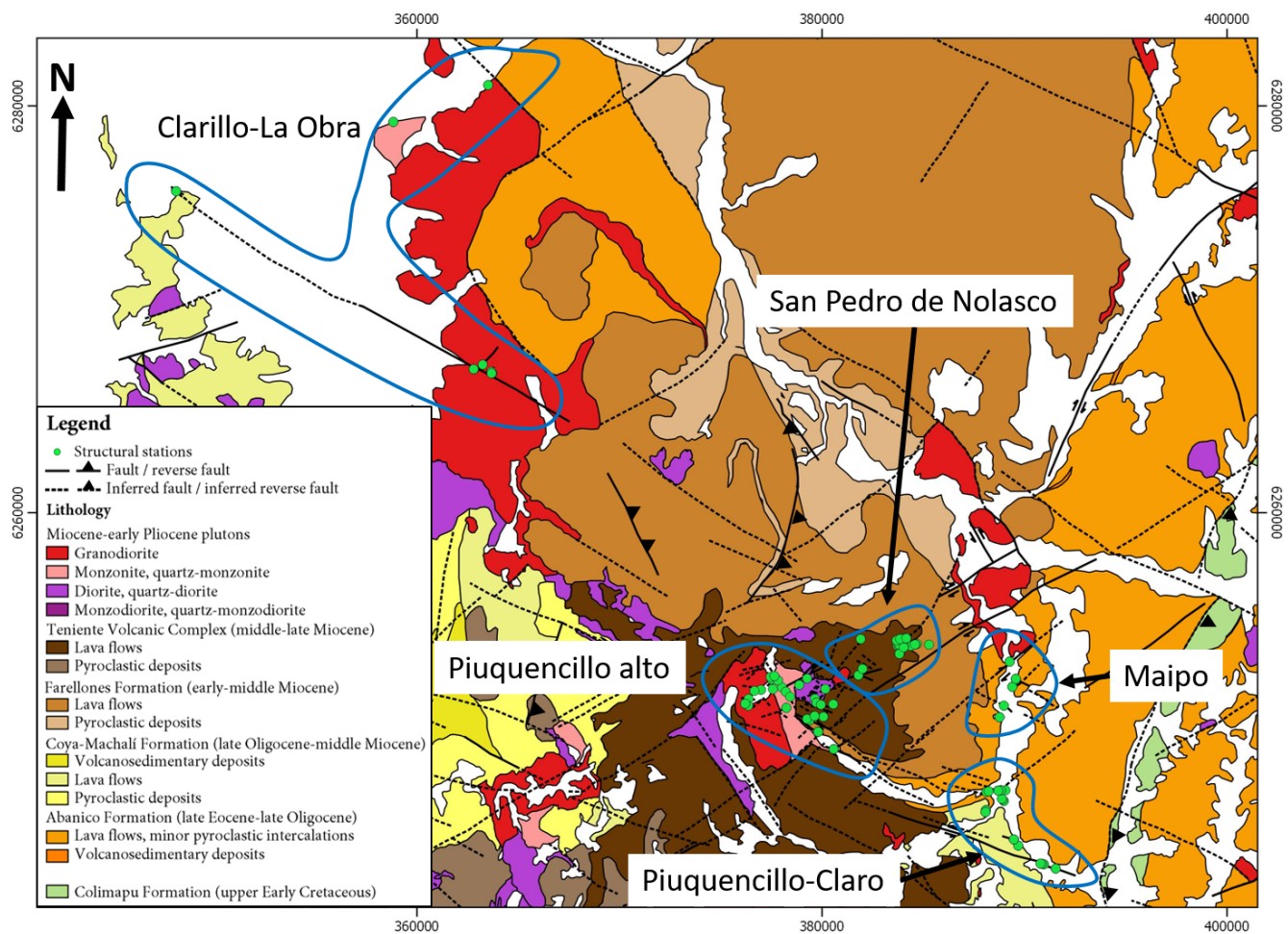

**Figure 5: Subdivision of the study area into five sectors for the analysis of preferred orientations and the calculation of strain axes and stress tensors. Geological map in the background compiled from Rivera and Cembrano (2000), SERNAGEOMIN (2002), Fock (2005), Fock et al. (2005), Piquer (2015) and this work.**

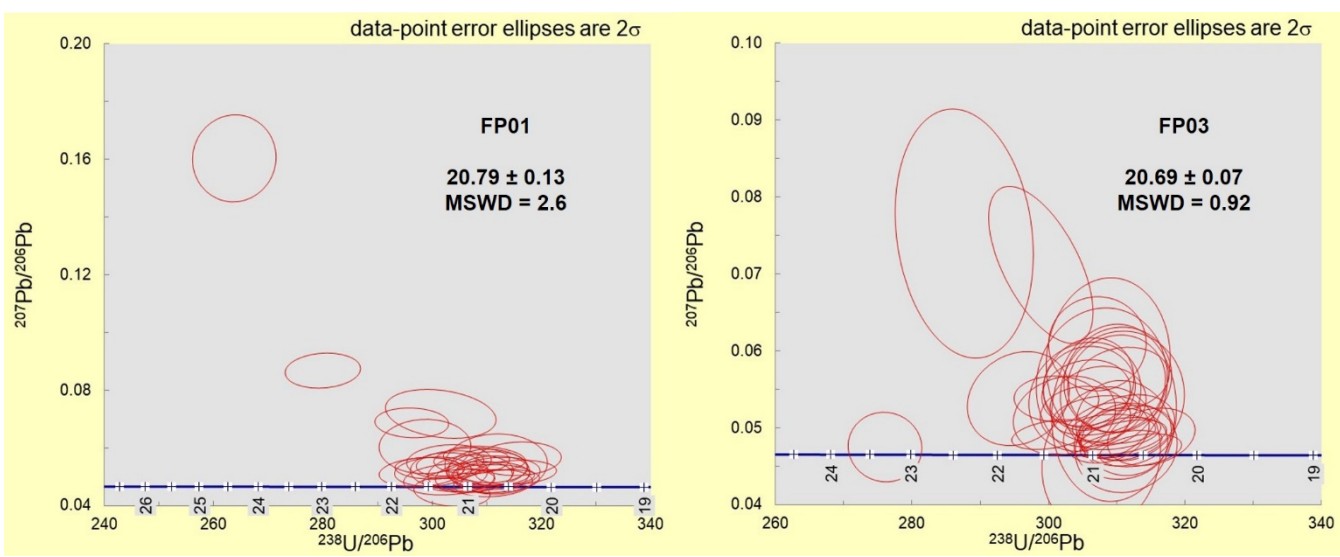

**Figure 6: Tera-Wasserburg plots for the two U-Pb analyses completed for this study. Numbers on the reference concordia traces are millions of years.**

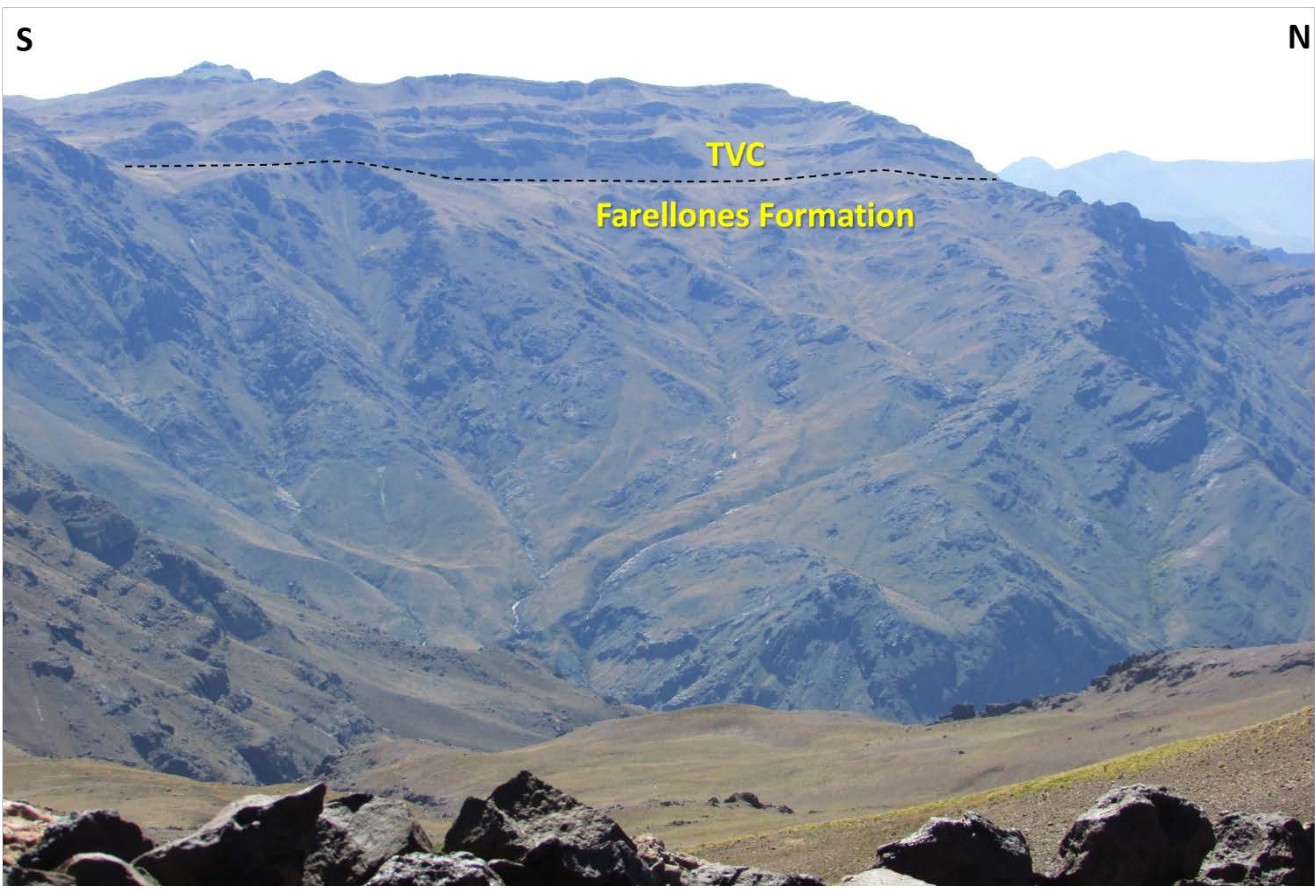


**Figure 7: Unconformity (dashed line) separating the Teniente Volcanic Complex (TVC) from the Farellones Formation. View west from the San Pedro de Nolasco vein area.**

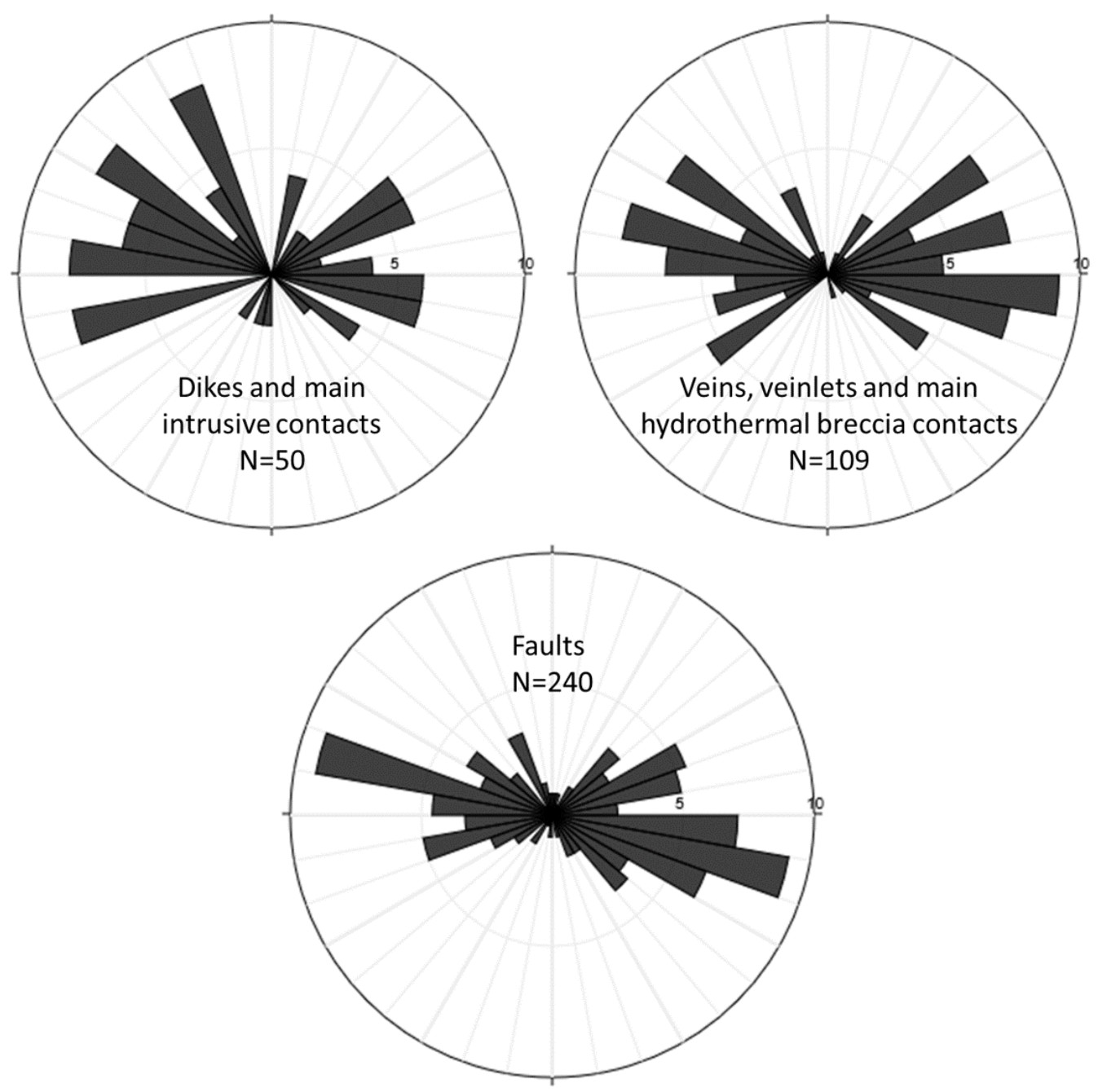

**Figure 8: Preferred orientations of dikes/main intrusive contacts, veins/main hydrothermal breccia contacts and faults, for all the study area (82 structural stations).**

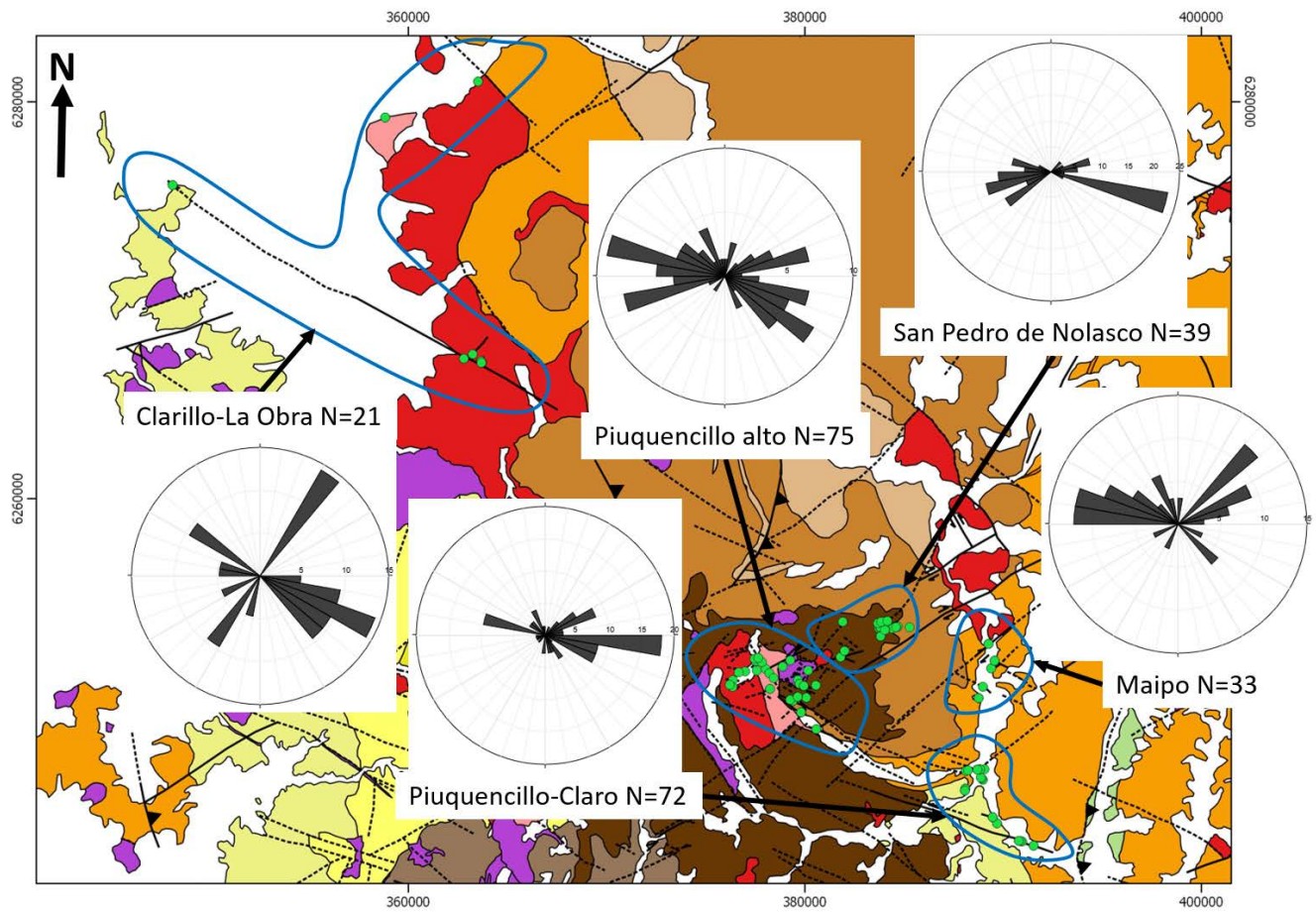

**Figure 9: Preferred orientations of fault planes in the five sectors into which the study area was subdivided. Geological map on the background as in Figure 5.**

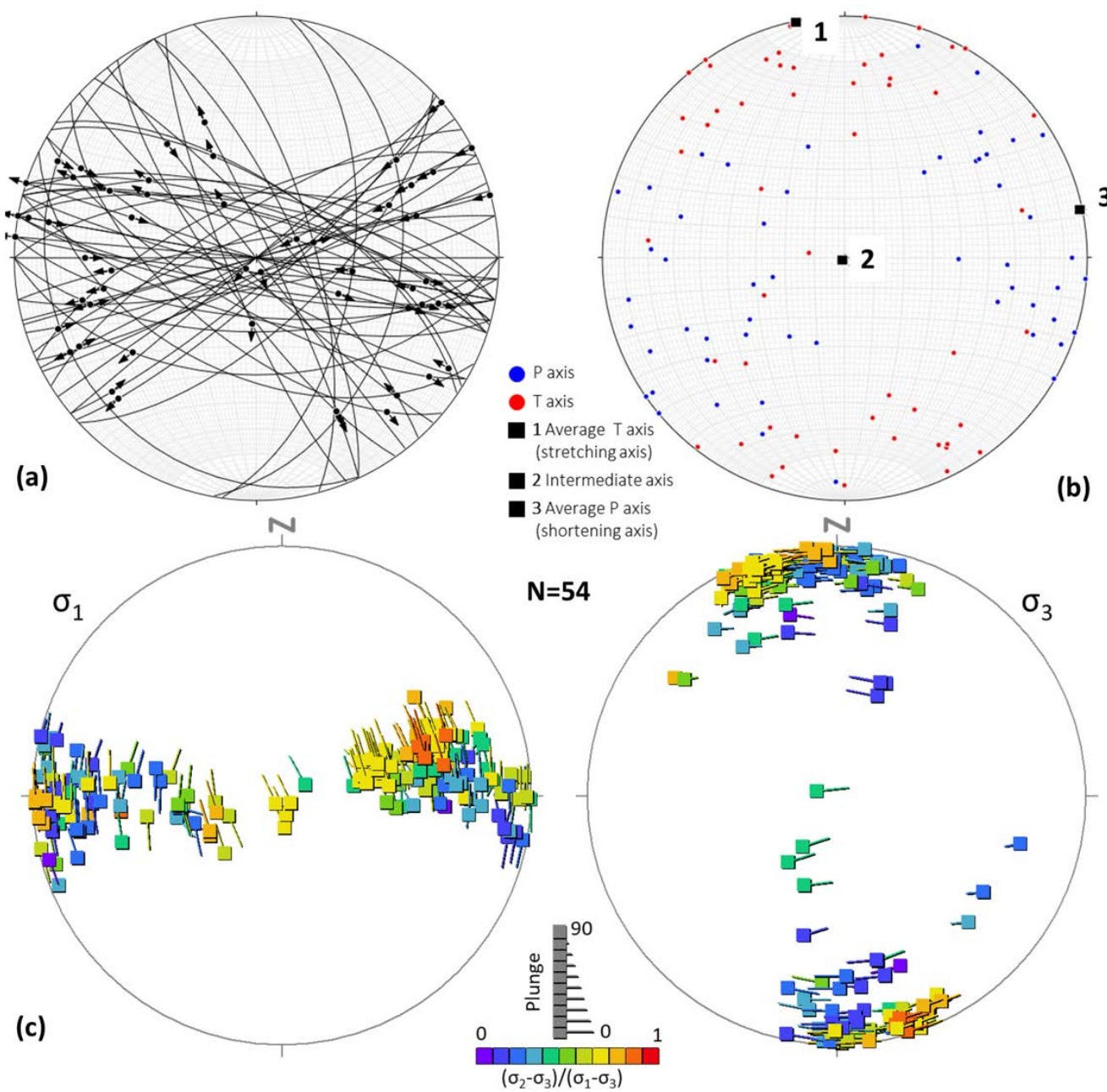

Figure 10: (a) Lower-hemisphere, equal-area projection of the 54 fault planes with kinematic information, showing also the slickenline attitudes and the sense of movement. (b) Results of the kinematic analysis for the fault planes shown in (a). P = pressure, T = tension. (c) Results of the dynamic analysis for the fault planes shown in (a). Colors represent the Φ value calculated for each stress tensor.

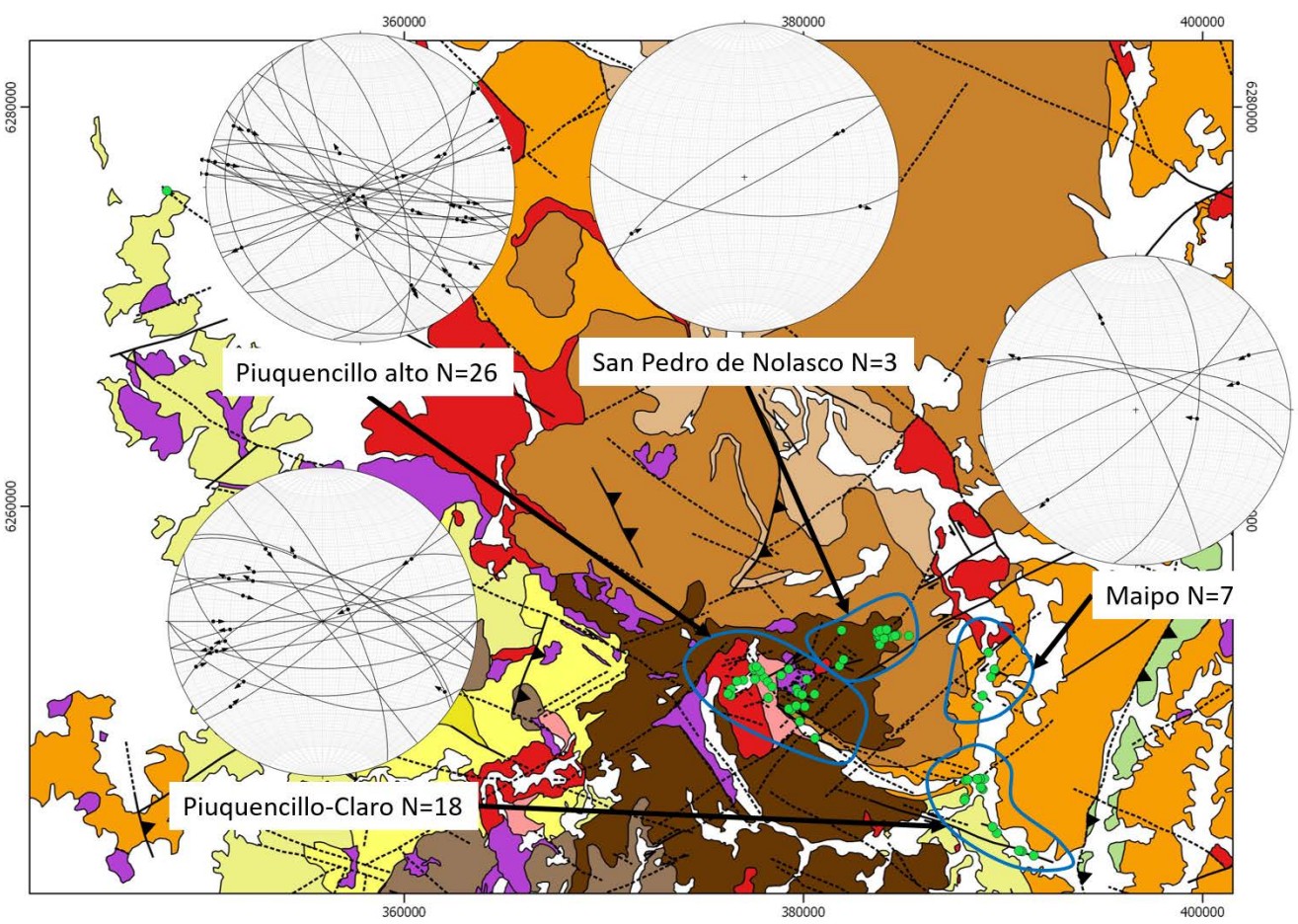

**Figure 11: Lower-hemisphere, equal-area projection of the fault planes with kinematic information by sector, showing also the slickenline attitudes and the sense of movement. Geological map on the background as in Figure 5.**

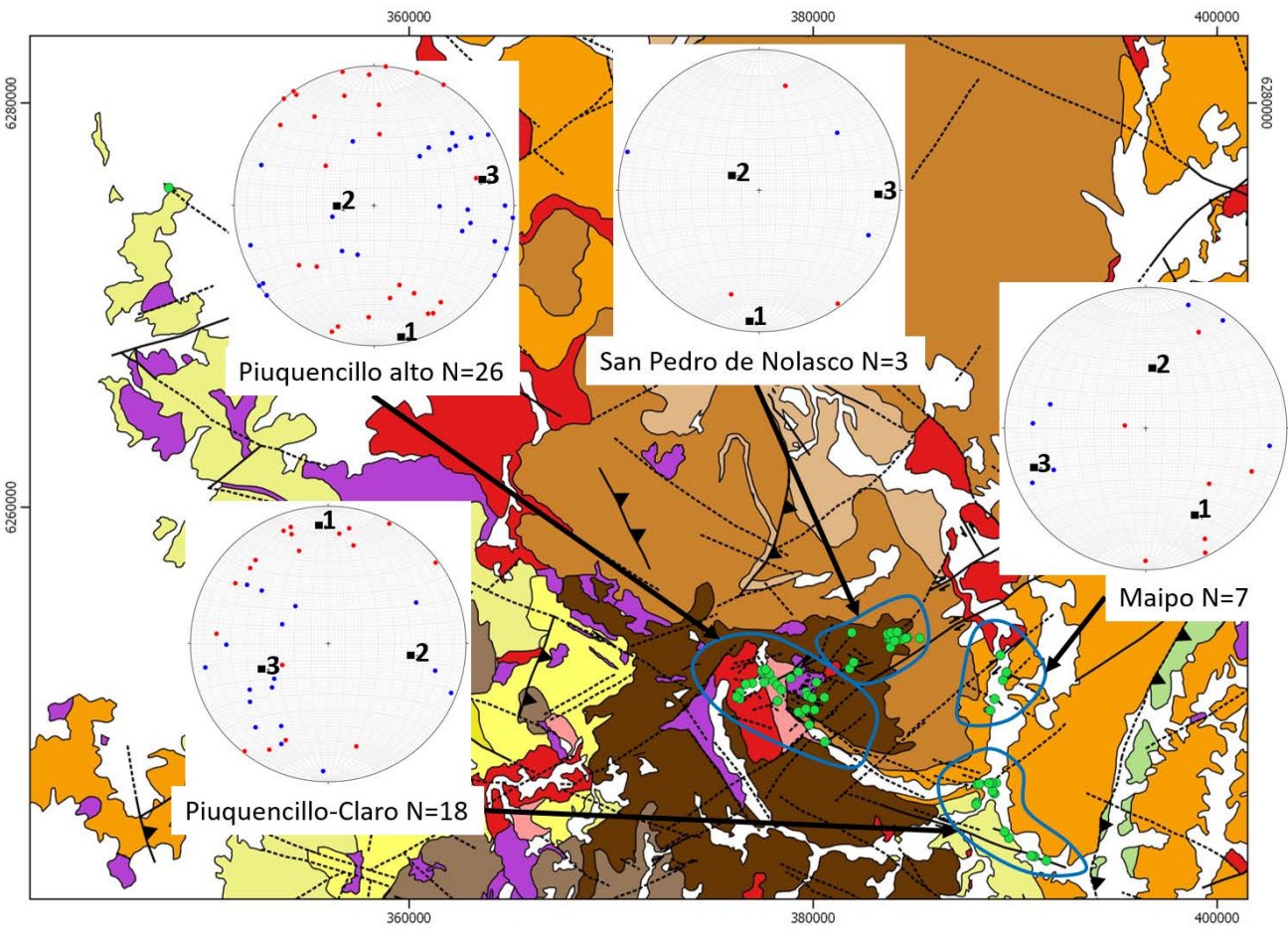

**Figure 12: Kinematic analysis of fault plane data by sector. Clarillo-La Obra sector is not considered, as no reliable kinematic indicators were found. Geological map on the background as in Figure 5.**

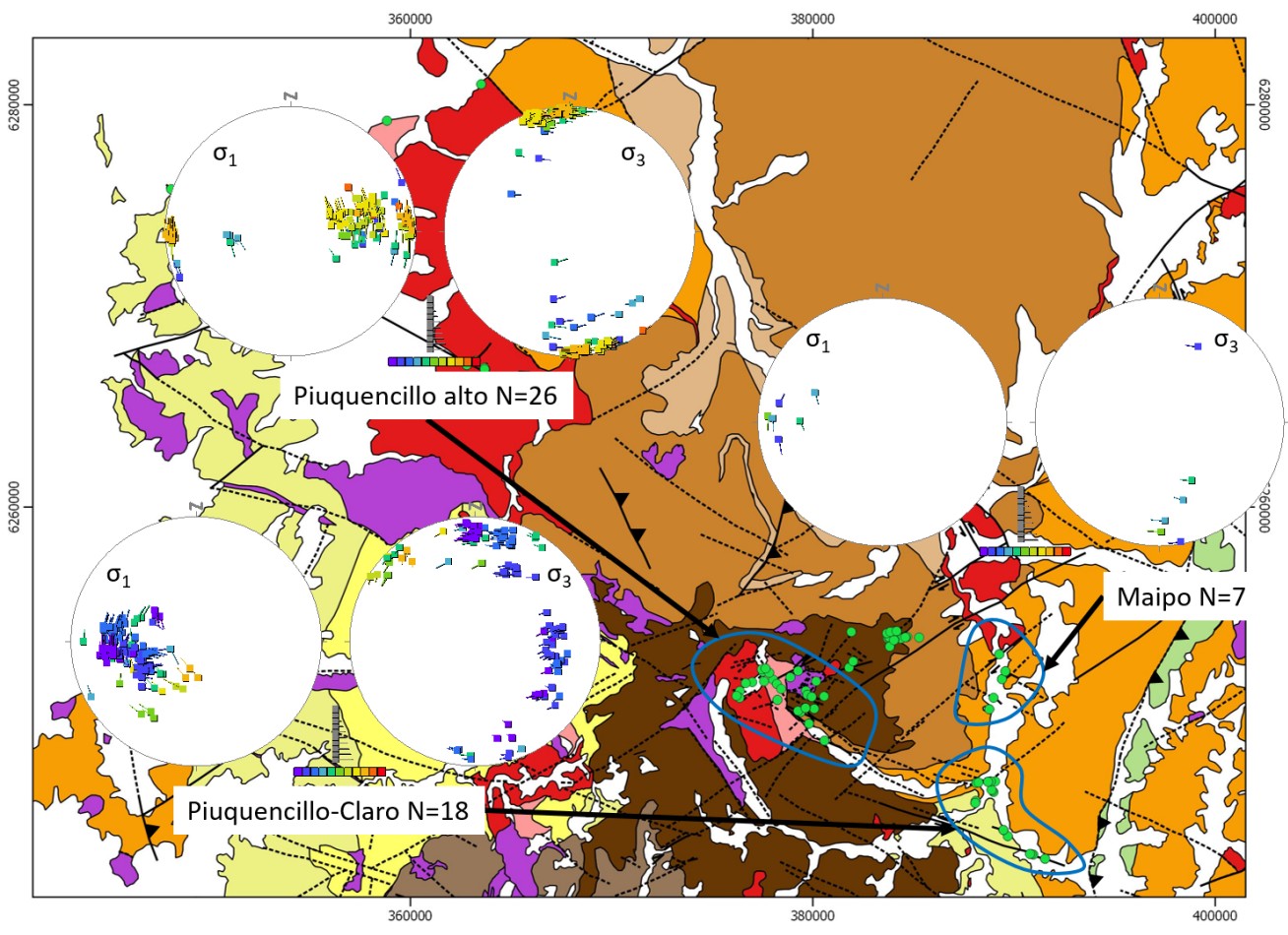

**Figure 13: Dynamic analysis of fault plane data by sector. Clarillo-La Obra and San Pedro de Nolasco sectors are not considered, as the minimum of four fault planes with reliable kinematic indicators, necessary for stress state calculations, was not achieved in these areas. Geological map on the background as in Figure 5.**

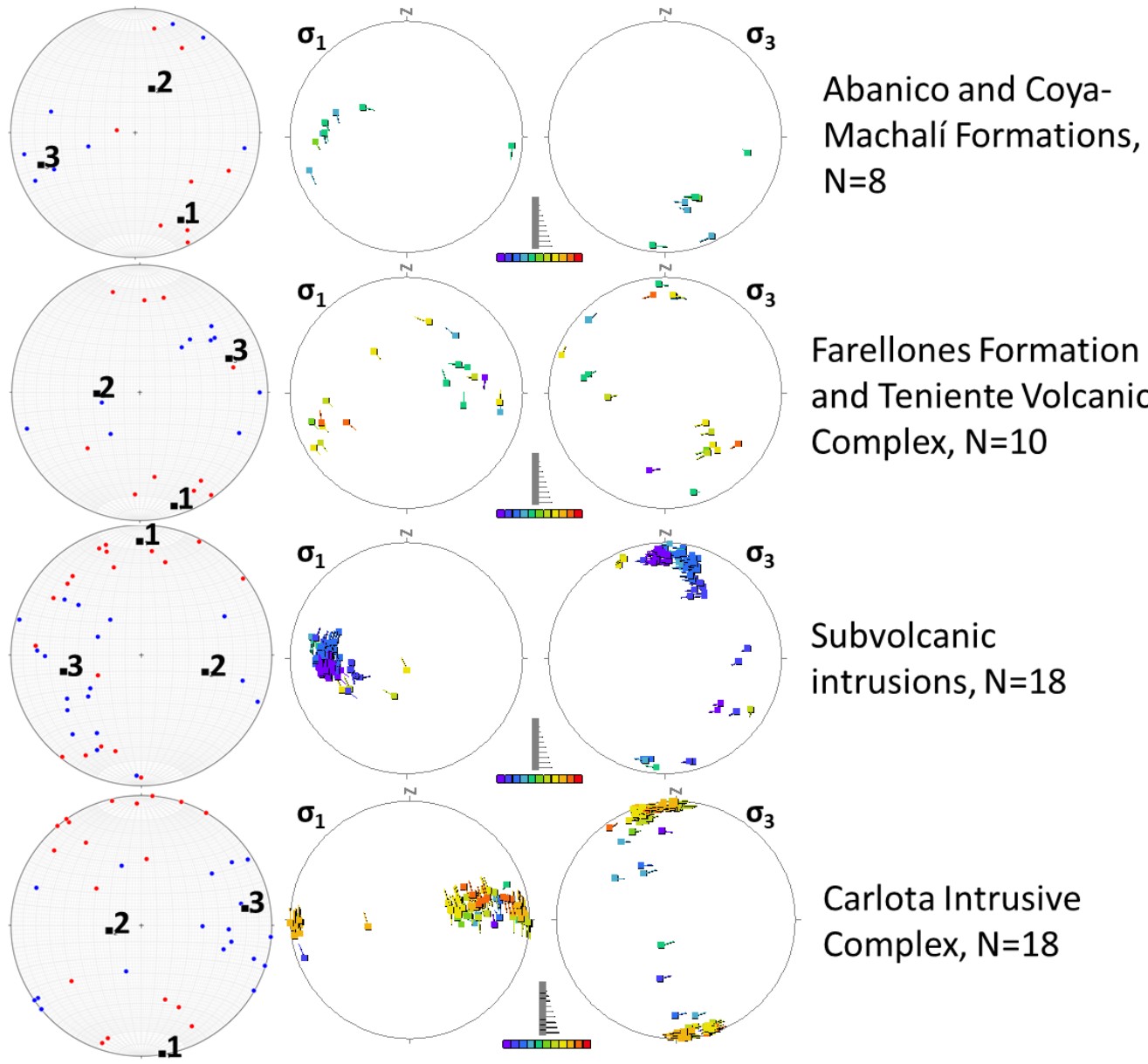


**Figure 14: Kinematic and dynamic analysis of fault plane data by lithological unit.**

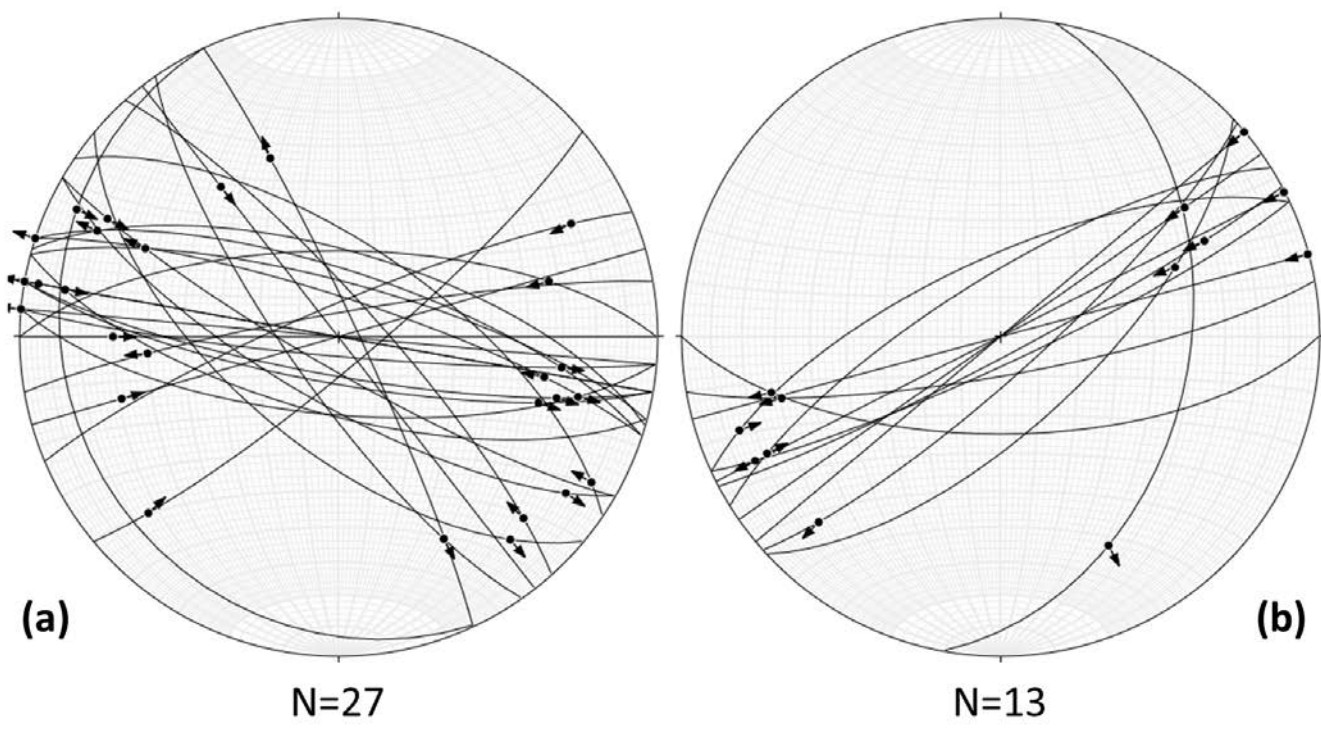

**Figure 15: (a) Lower-hemisphere, equal-area projection of fault planes with a sinistral strike-slip component and striations with pitch ≤45°. (b) Same as in (a) but for fault planes with a dextral slip.**

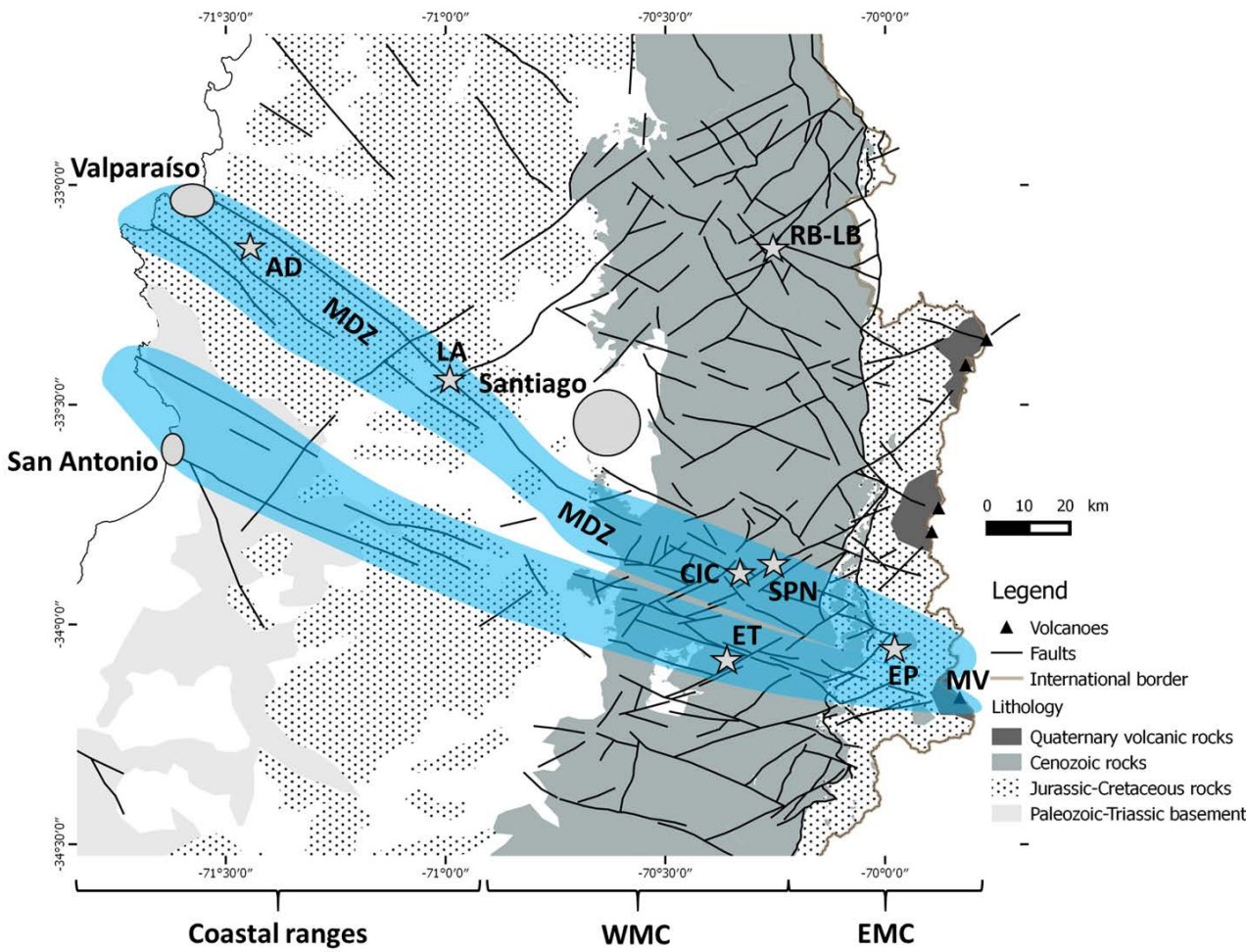


**Figure 16: Continental-scale expression of the Maipo deformation zone (MDZ), highlighted in light blue; the PFS correspond to its manifestation in the Cenozoic rocks of the Western Main Cordillera. Background geology from SERNAGEOMIN (2002); faults in the Western Main Cordillera from Piquer et al. (2016). RB-LB = Río Blanco-Los Bronces; ET = El Teniente; EP = Escalones prospect; SPN = San Pedro de Nolasco veins; CIC = Carlota Intrusive Complex; LA = Lo Aguirre stratabound deposit; AD = Antena District of orogenic Au veins; MV = Maipo Volcano; WMC = Western Main Cordillera; EMC = Eastern Main Cordillera. The main cities of Santiago and Valparaíso are also shown.**


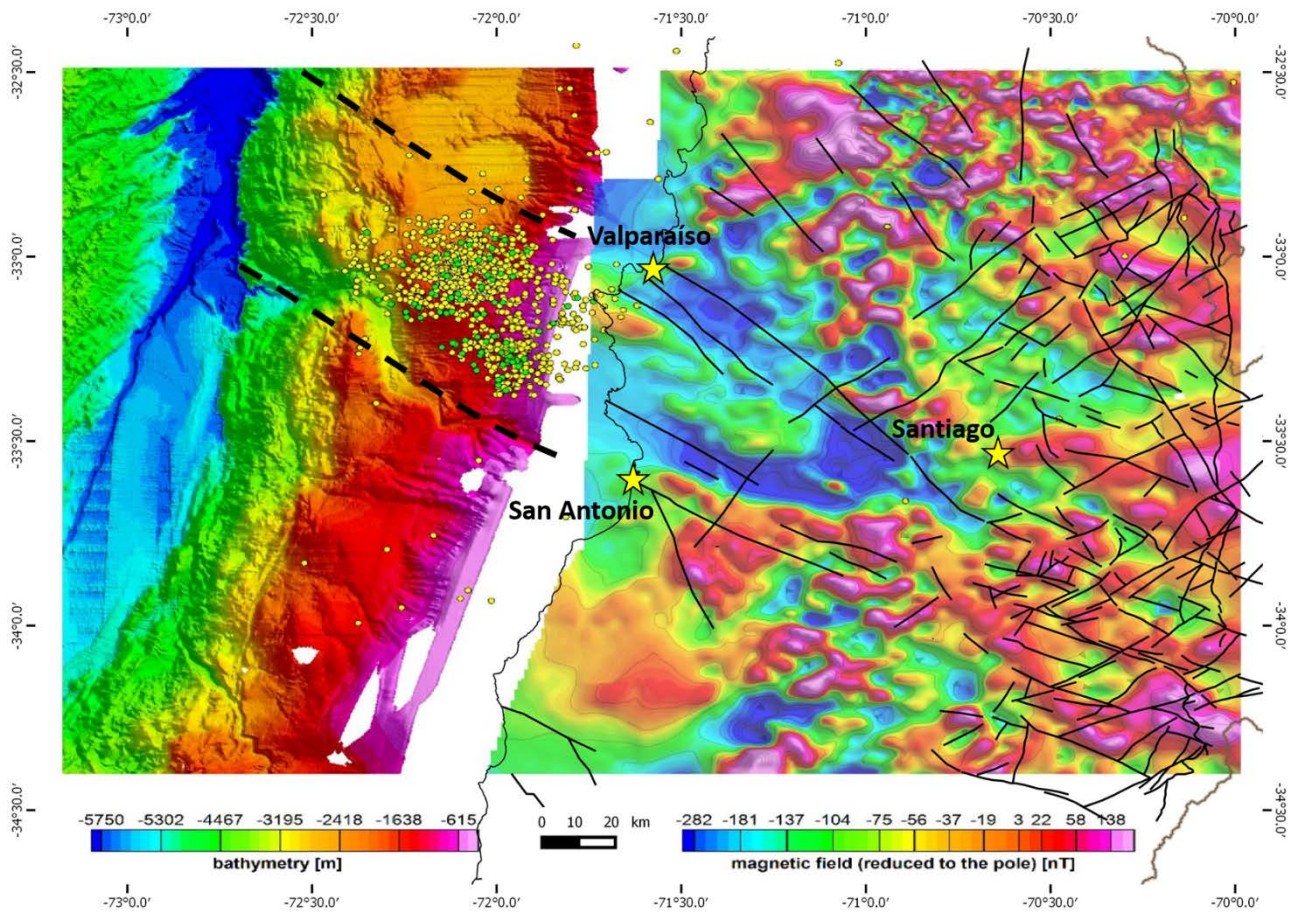

**Figure 17: Geophysical expressions of the PFS, the Maipo Deformation Zone and the Teniente-San Antonio fault system. The submarine continental shelf is colored according to its bathymetry (Weinrebe and Hasert, 2012); thick dashed lines represent the possible submarine prolongation of the two main branches of the MDZ. Colors in the continent correspond to RTP aeromagnetic map (SERNAGEOMIN, 1980). Yellow and green small circles show the location of earthquake hypocenters related to the 2017 Valparaíso sequence (yellow data set extracted from CSN data repository, green data set, relocated events from Nealy et al 2017, data repository). The Chile-Argentina international boundary, the coastline and fault architecture from Fig. 16 are also shown.**

| Sample | N (UTM) | E (UTM) | Lithology | Age (Ma) ($\pm 2\sigma$) | MSWD | Number of grains |
|---|---|---|---|---|---|---|
| FP01 | 6281776 | 361704 | Biotite-rich granodiorite | $20.79 \pm 0.13$ | 2.6 | 29 |
| FP03 | 6279214 | 358835 | Hornblende-rich monzogranite | $20.69 \pm 0.07$ | 0.92 | 34 |

**Table 1: Summary of U-Pb zircon geochronology of the La Obra batholith.**

**Appendix A: U-Pb geochronology analytical procedures**

All the procedures described here were completed at the Geochronology Laboratory of SERNAGEOMIN (National Survey of Geology and Mining, Chile) at Santiago, Chile. First the samples were sieved and crushed using standard procedures to obtain a non-magnetic heavy mineral concentrate, from which zircon crystals were separated. The selected crystals were mounted in epoxy glue briquettes, and then studied at a Scanning Electron Microscope, with which back-scattered electron (BSE) and cathodoluminescence (CL) images were obtained.

At sample FP01, the selected zircon crystals vary in size between 100 and 300 μm approximately. They correspond to igneous zircons with oscillatory zoning, some of them showing evidences of inherited cores. No overgrowth was observed at crystal rims. The grains contain abundant inclusions and fractures, which required a careful selection of the laser ablation spots.

At sample FP03, the selected zircon crystals vary in size between 100 and 500 μm approximately. They correspond to igneous zircons with oscillatory zoning, without observed evidences of inherited cores. Overgrowth was observed at the rims of some crystals, and some of them display irregular borders, maybe as a result of hydrothermal processes. The grains contain very few inclusions and fractures.

After spot selection based on CL and BSE images, the analyses were completed using a Thermo Fischer ElementXR ICP-MS, coupled with a Photon-Machines Analyte G2 193 nm excimer laser, with a wavelength of 193 nm. U, Th and Pb concentrations were calculated in relation to the reference zircon GJ-1 (Jackson et al., 2004). Isotope ratios were normalized to the same reference zircon.

## Appendix B: U-Pb geochronology analytical data

Values in bold correspond to those used for age calculations.

(1) Values not considered in the calculations due to an incorrect pattern in isotopic ratio curves (isotopic content inhomogeneity)

700

| $^{206}Pb/^{238}U$ (corr by common Pb) | | $^{206}Pb/^{238}U$ | | $^{207}Pb/^{235}U$ | | | $^{207}Pb/^{206}Pb$ | | $^{207}Pb/^{206}Pb$ |
|---|---|---|---|---|---|---|---|---|---|
| age | 2σ | ratio | 2σ | ratio | 2σ | Rho | ratio | 2σ | Common |
| **FP01** | | | | | | | | | |
| **21.2** | **0.5** | 0.00330 | 0.00007 | 0.02233 | 0.00200 | 0.07485 | 0.04940 | 0.00450 | 0.83698 |
| **20.8** | **0.5** | 0.00326 | 0.00007 | 0.02291 | 0.00240 | 0.07219 | 0.05170 | 0.00520 | 0.83696 |
| **21.1** | **0.5** | 0.00335 | 0.00008 | 0.02791 | 0.00360 | 0.15235 | 0.06120 | 0.00730 | 0.83700 |
| **21.0** | **0.4** | 0.00330 | 0.00006 | 0.02483 | 0.00160 | 0.02329 | 0.05440 | 0.00340 | 0.83698 |
| **21.0** | **0.4** | 0.00328 | 0.00006 | 0.02204 | 0.00140 | 0.11814 | 0.04890 | 0.00300 | 0.83697 |
| **20.6** | **0.4** | 0.00322 | 0.00006 | 0.02310 | 0.00280 | 0.18976 | 0.05240 | 0.00600 | 0.83695 |
| **20.6** | **0.4** | 0.00323 | 0.00006 | 0.02339 | 0.00220 | 0.05477 | 0.05340 | 0.00490 | 0.83695 |
| **20.4** | **0.5** | 0.00321 | 0.00008 | 0.02406 | 0.00380 | 0.11754 | 0.05690 | 0.00820 | 0.83694 |
| **21.1** | **0.4** | 0.00338 | 0.00006 | 0.03196 | 0.00210 | 0.35555 | 0.06820 | 0.00410 | 0.83701 |
| **21.2** | **0.4** | 0.00329 | 0.00006 | 0.02031 | 0.00200 | 0.36009 | 0.04520 | 0.00410 | 0.83698 |
| **21.3** | **0.4** | 0.00334 | 0.00006 | 0.02435 | 0.00150 | 0.35256 | 0.05280 | 0.00270 | 0.83699 |
| **20.6** | **0.4** | 0.00321 | 0.00006 | 0.02243 | 0.00140 | 0.34455 | 0.05060 | 0.00280 | 0.83694 |
| **20.9** | **0.6** | 0.00379 | 0.00009 | 0.08384 | 0.00670 | 0.27281 | 0.16000 | 0.01100 | 0.83718 |
| **20.5** | **0.4** | 0.00321 | 0.00007 | 0.02262 | 0.00200 | 0.08329 | 0.05180 | 0.00450 | 0.83694 |
| **21.3** | **0.4** | 0.00333 | 0.00006 | 0.02377 | 0.00190 | 0.10277 | 0.05170 | 0.00400 | 0.83699 |
| **20.5** | **0.4** | 0.00321 | 0.00006 | 0.02262 | 0.00150 | 0.25837 | 0.05160 | 0.00310 | 0.83694 |
| **20.5** | **0.4** | 0.00320 | 0.00006 | 0.02137 | 0.00140 | 0.00231 | 0.04900 | 0.00310 | 0.83694 |
| **21.4** | **0.5** | 0.00336 | 0.00007 | 0.02387 | 0.00190 | 0.08780 | 0.05300 | 0.00410 | 0.83700 |
| **20.7** | **0.4** | 0.00325 | 0.00006 | 0.02493 | 0.00170 | 0.20271 | 0.05640 | 0.00360 | 0.83696 |
| **20.5** | **0.4** | 0.00322 | 0.00006 | 0.02329 | 0.00170 | 0.13290 | 0.05320 | 0.00340 | 0.83694 |
| **20.6** | **0.6** | 0.00332 | 0.00009 | 0.03282 | 0.00350 | 0.48085 | 0.07440 | 0.00730 | 0.83698 |
| **20.5** | **0.4** | 0.00323 | 0.00006 | 0.02512 | 0.00170 | 0.47512 | 0.05790 | 0.00340 | 0.83695 |

| age | 2σ | ratio | 2σ | ratio | 2σ | Rho | ratio | 2σ | Common |
|---|---|---|---|---|---|---|---|---|---|
| **21.8** | **0.5** | 0.00357 | 0.00007 | 0.04274 | 0.00250 | 0.23171 | 0.08650 | 0.00450 | 0.83709 |
| **21.0** | **0.5** | 0.00330 | 0.00007 | 0.02512 | 0.00220 | 0.14942 | 0.05600 | 0.00450 | 0.83698 |
| **20.8** | **0.4** | 0.00325 | 0.00006 | 0.02243 | 0.00190 | 0.04361 | 0.05160 | 0.00440 | 0.83696 |
| **20.6** | **0.4** | 0.00321 | 0.00006 | 0.02118 | 0.00170 | 0.07506 | 0.04910 | 0.00380 | 0.83694 |
| **21.1** | **0.4** | 0.00330 | 0.00006 | 0.02368 | 0.00240 | 0.21186 | 0.05320 | 0.00550 | 0.83698 |
| **20.3** | **0.4** | 0.00317 | 0.00006 | 0.02252 | 0.00150 | 0.14284 | 0.05220 | 0.00330 | 0.83692 |
| **20.1** | **0.4** | 0.00317 | 0.00006 | 0.02454 | 0.00210 | 0.14760 | 0.05650 | 0.00470 | 0.83692 |
| (1) | | ~~0.00387~~ | ~~0.00019~~ | ~~0.07508~~ | ~~0.02200~~ | ~~0.99260~~ | ~~0.11100~~ | ~~0.01800~~ | |
| (1) | | ~~0.00434~~ | ~~0.00011~~ | ~~0.15304~~ | ~~0.01700~~ | ~~0.11226~~ | ~~0.25300~~ | ~~0.02700~~ | |
| (1) | | ~~0.00378~~ | ~~0.00011~~ | ~~0.07912~~ | ~~0.00560~~ | ~~0.13594~~ | ~~0.15460~~ | ~~0.00980~~ | |
| (1) | | ~~0.00437~~ | ~~0.00031~~ | ~~0.15016~~ | ~~0.03100~~ | ~~0.99093~~ | ~~0.21300~~ | ~~0.03000~~ | |
| (1) | | ~~0.11283~~ | ~~0.02400~~ | ~~12.51304~~ | ~~2.80000~~ | ~~0.99880~~ | ~~0.78600~~ | ~~0.01100~~ | |

| $^{206}Pb/^{238}U$ (corr by common Pb) | | $^{206}Pb/^{238}U$ | | $^{207}Pb/^{235}U$ | | | $^{207}Pb/^{206}Pb$ | | $^{207}Pb/^{206}Pb$ |
|---|---|---|---|---|---|---|---|---|---|
| age | 2σ | ratio | 2σ | ratio | 2σ | Rho | ratio | 2σ | Common |
| **FP03** | | | | | | | | | |
| **21.2** | **0.4** | 0.00332 | 0.00006 | 0.02461 | 0.00120 | 0.37242 | 0.05350 | 0.00240 | 0.83699 |
| **20.8** | **0.4** | 0.00327 | 0.00006 | 0.02461 | 0.00160 | 0.14300 | 0.05490 | 0.00330 | 0.83697 |
| **20.8** | **0.4** | 0.00324 | 0.00005 | 0.02206 | 0.00140 | 0.01562 | 0.04990 | 0.00310 | 0.83695 |
| **20.5** | **0.5** | 0.00324 | 0.00008 | 0.02617 | 0.00400 | 0.11329 | 0.05880 | 0.00860 | 0.83695 |
| **20.7** | **0.4** | 0.00325 | 0.00006 | 0.02382 | 0.00180 | 0.11766 | 0.05460 | 0.00400 | 0.83696 |
| **20.5** | **0.4** | 0.00326 | 0.00006 | 0.02735 | 0.00230 | 0.21824 | 0.06300 | 0.00500 | 0.83696 |
| **20.6** | **0.4** | 0.00321 | 0.00006 | 0.02098 | 0.00210 | 0.09036 | 0.04790 | 0.00480 | 0.83694 |
| **21.6** | **0.5** | 0.00338 | 0.00007 | 0.02510 | 0.00230 | 0.05167 | 0.05220 | 0.00450 | 0.83701 |
| **21.6** | **0.7** | 0.00348 | 0.00010 | 0.03608 | 0.00660 | 0.31855 | 0.07500 | 0.01300 | 0.83705 |
| **20.5** | **0.5** | 0.00322 | 0.00007 | 0.02500 | 0.00250 | 0.12196 | 0.05680 | 0.00570 | 0.83695 |
| **21.1** | **0.4** | 0.00331 | 0.00007 | 0.02353 | 0.00190 | 0.08736 | 0.05190 | 0.00420 | 0.83698 |
| **20.6** | **0.4** | 0.00322 | 0.00006 | 0.02216 | 0.00230 | 0.23467 | 0.04990 | 0.00520 | 0.83694 |
| **20.5** | **0.5** | 0.00322 | 0.00008 | 0.02402 | 0.00310 | 0.16945 | 0.05490 | 0.00710 | 0.83694 |

| 20.9 | 0.5 | 0.00325 | 0.00008 | 0.02059 | 0.00340 | 0.00612 | 0.04550 | 0.00740 | 0.83696 |
| 20.7 | 0.3 | 0.00322 | 0.00005 | 0.02124 | 0.00058 | 0.06152 | 0.04790 | 0.00120 | 0.83694 |
| 20.5 | 0.4 | 0.00323 | 0.00006 | 0.02480 | 0.00230 | 0.11619 | 0.05540 | 0.00520 | 0.83695 |
| 20.5 | 0.4 | 0.00322 | 0.00006 | 0.02500 | 0.00150 | 0.16404 | 0.05520 | 0.00340 | 0.83694 |
| 20.8 | 0.4 | 0.00327 | 0.00006 | 0.02510 | 0.00220 | 0.07492 | 0.05570 | 0.00460 | 0.83697 |
| 20.5 | 0.3 | 0.00319 | 0.00005 | 0.02070 | 0.00073 | 0.09545 | 0.04770 | 0.00170 | 0.83693 |
| 20.4 | 0.3 | 0.00317 | 0.00005 | 0.02137 | 0.00130 | 0.03120 | 0.04900 | 0.00290 | 0.83693 |
| 20.7 | 0.3 | 0.00322 | 0.00005 | 0.02188 | 0.00071 | 0.09875 | 0.04910 | 0.00160 | 0.83695 |
| 20.6 | 0.6 | 0.00325 | 0.00008 | 0.02500 | 0.00360 | 0.11666 | 0.05670 | 0.00780 | 0.83696 |
| 21.3 | 0.3 | 0.00333 | 0.00005 | 0.02255 | 0.00068 | 0.12092 | 0.04930 | 0.00140 | 0.83699 |
| 20.6 | 0.4 | 0.00321 | 0.00006 | 0.02216 | 0.00150 | 0.02902 | 0.04970 | 0.00320 | 0.83694 |
| 20.8 | 0.4 | 0.00326 | 0.00006 | 0.02317 | 0.00092 | 0.15464 | 0.05220 | 0.00210 | 0.83696 |
| 20.6 | 0.3 | 0.00322 | 0.00005 | 0.02163 | 0.00074 | 0.01478 | 0.04890 | 0.00160 | 0.83694 |
| 20.6 | 0.6 | 0.00323 | 0.00008 | 0.02314 | 0.00420 | 0.07568 | 0.05300 | 0.01000 | 0.83695 |
| 20.8 | 0.5 | 0.00328 | 0.00007 | 0.02441 | 0.00260 | 0.06892 | 0.05510 | 0.00590 | 0.83697 |
| 20.8 | 0.4 | 0.00327 | 0.00006 | 0.02490 | 0.00200 | 0.18788 | 0.05500 | 0.00440 | 0.83697 |
| 20.9 | 0.4 | 0.00327 | 0.00006 | 0.02323 | 0.00150 | 0.25362 | 0.05170 | 0.00320 | 0.83697 |
| 20.9 | 0.5 | 0.00335 | 0.00007 | 0.03284 | 0.00430 | 0.70012 | 0.07090 | 0.00800 | 0.83700 |
| 20.6 | 0.4 | 0.00321 | 0.00006 | 0.02167 | 0.00140 | 0.07098 | 0.04920 | 0.00320 | 0.83694 |
| 20.7 | 0.3 | 0.00323 | 0.00005 | 0.02220 | 0.00077 | 0.43799 | 0.05020 | 0.00150 | 0.83695 |
| 20.7 | 0.4 | 0.00322 | 0.00006 | 0.02127 | 0.00170 | 0.16802 | 0.04790 | 0.00360 | 0.83694 |
| 23.3 | 0.4 | 0.00362 | 0.00006 | 0.02372 | 0.00190 | 0.23066 | 0.04750 | 0.00360 | 0.83711 |
| (1) | | 0.00352 | 0.00009 | 0.05049 | 0.00650 | 0.53069 | 0.10300 | 0.01200 | |
| (1) | | 0.00343 | 0.00010 | 0.05000 | 0.01000 | 0.91903 | 0.09400 | 0.01400 | |
| (1) | | 0.00262 | 0.00006 | 0.01760 | 0.00093 | 0.06173 | 0.04910 | 0.00270 | |
| (1) | | 0.00373 | 0.00007 | 0.03157 | 0.00200 | 0.07834 | 0.06150 | 0.00390 | |