# Peer review of "The Piuquencillo Fault System: a long-lived, Andean-transverse fault system and its relationship with magmatic and hydrothermal activity"

_Solid Earth, 2020_

## Referee Comment (RC1) · Laura Giambiagi (Referee) · 9 Sep 2020

The paper by Piquer and coauthors, on the structural characterization of a NW-striking fault system in the southern Central Andes, reports new fault kinematic data and performs stress inversion analysis, which are used to discuss the relationship between pre-existing structures, local stress field, and fluid migration. The results from these analyses are then integrated with geophysical data to provide an integral characterization of the fault system. The paper is well-written, and of wide interest to the broad Solid Earth audience. Below, I provide six recommendations on how to clarify and enhance the manuscript, to make it a stronger contribution: (1) Tectonic setting: To give

the manuscript a broader impact, I suggest adding a tectonic setting section, which can be integrated into the Geological-tectonic setting, with a synthesis of the Cenozoic extensional and compressional phases, and a description of the main structures of the area. Also, I would add a description of the Miocene intrusive complexes, such as the La Obra batholith. (2) Lithospheric-scale structures: Since there is no evidence that these fault systems involve the continental lithosphere, it is more convenient to name them as "continental-scale structural systems" (as they are called in the Discussion section). An alternative approach could be to use the available geophysical data and clearly propose the connection between the crustal structures and deeply-seated anisotropies. These geophysical data/analyses are just mentioned, but not properly presented as a discussion. (3) Kinematic vs dynamic analysis of fault-slip data: This must be more deeply discussed. Why do the authors perform these two kinds of analyses? What are the reduced stress tensors obtained from fault-slip data close to a major long-lived trans-crustal fault system telling us in terms of dynamics? Can they be interpreted as reflecting the stress of the crust during the movement of these structures? Since these structures have been previously generated during at least the Oligocene extension, or probably before (as the authors mention in the Discussion), these tensors are not probably reflecting a stress field because, in this case, one of the basic assumptions during the inversion technique, the one concerning the absence of interference between faults, is not properly fulfilled. I suggest that the authors discriminate between stations close to the main faults, from stations located far away from the main faults. For example, data from the Piuquencillo Alto area could be separated into two (close and apart from the NW fault). (4) Age of dikes and intrusives: You should clarify the relationship between the emplacement of dikes and intrusives, and movement along the study fault system. Are there any time constraints on this relationship? For example, what is the age of the Ag-Pb-Zn-Cu vein system in the San Pedro de Nolasco sector? In this respect, move paragraph 182-186 from the Discussion section to the Results section, and add any other information about the timing of emplacement of these intrusives and the timing of slips of the studied faults. (5) The fluid pathways.:

This idea must be broadened. Discuss here how sigma 1 is obtained. Are WNW and ENE conjugated strike-slip faults, or not? For each measurement station, add the main strike of dextral and sinistral faults, as well as the orientation of local stress tensor axes, and then compare them. As I pointed out above, it would be nice to have had previously discrimination between tensors obtained close to or far apart from the main faults. It is not clear to me why the ENE striking faults are optimally-oriented for the migration of fluids, while the WNW striking faults are not. If both sets are conjugate strike-slip faults, as stated in lines 201-202, why is one set more likely to dilate and the other one more likely to slip? In my opinion, more information about the strike of the local sigma 1 is needed to evaluate this proposal. Add a table containing: location of each station, number of measured faults, results of fault-slip inversion (orientation of principal stress axes and stress ratio), and lapse of time for each calculated reduced stress tensor. (6) Transient stress reversals: This statement is very questionable. It is difficult to explain a change from compression to extension as stress reversals during the coseismic stress release. Positive changes in the Coulomb failure stress (see Harris et al. 1998; Stein, 1999; King and Cocco, 2000; Freed 2005) bring receiver faults in the continental crust closer to failure, but the magnitudes involved in these stress changes are small (<1 MPa). The process is more related to the unclampling of the receiver faults, due to a drop in the normal stress that prevents the slip during the interseismic period. See Spagnotto et al (2015), who explore how changes in the Coulomb conditions associated with the Maule earthquake triggered upper-plate earthquakes. They argue that the reactivation of faults after a mega-earthquake at the subduction interphase derives from unclamping processes associated with co-seismic dilatation deformation inferred from GPS observations. To summarise, I recommend that this important paper for the understanding of the relationship between pre-existing structures, local stress field and fluid migration should be published in Solid Earth, after some important points are addressed to make it an even stronger contribution.

Sincerely, Laura Giambiagi

Minor comments are listed below: Line 59: contractional deformation instead of compressive deformation Line 64: by whom? Line 75: Here I wonder how are these sheeted dikes related to mineralization in the El Teniente porphyry? Line 80: The term "progressive unconformities" is used to describe unconformities between beds of a single stratigraphical unit, not between two units such as Abanico and Farellones Fms. Lines 85-87: This suggests that the crust was thicker in the north than in the south during the extrusion of these units, but it does not necessarily imply that Cenozoic compression has started first in the north. The crust may have been thickened during previous compressional phases or it may not have been extended during the Abanico extension as much as the southern segment. Line 103: Reduced paleo-stress tensors: Briefly describe here which is the methodology to separate heterogeneous data sets into homogeneous ones, since this is a very critical step to obtained robust tensors. Line 115: Is the intrusion of the La Obra batholith related to the NW striking fault system? Stress here the importance of the new dating. Line 144: Can these dikes, intrusives and breccia be more constrained in age? Lines 174-176: In a strike-slip fault system, fault-slip sets taken from stations close to the system may have been formed under extensional and compressional local stress fields, generated a bend of faults, step overs, etc. Line 178: Explain how the ENE trend of sigma1 has been obtained. Lines 191-192: Discuss the possibility that the change from compressive in the highest sectors to compressive/strike-slip stress regime in the lowest sectors could also be related to excess of gravitational potential energy in the highest sectors of the Principal Cordillera. Lines 244-250. Part of this structural information may be move to the tectonic or structural setting. Lines 277-278: The focal mechanism solution for the Las Melosas earthquake is quite particular, with NNW-oriented P axis, not compatible with the WNW-striking sinistral faults of the PFS. Figure 1: add sense of displacement of normal and strike-slip faults Figure 2: Improve the location map (a) to show the main tectonic characteristics of the study area. Add lat- long in the map from Figure 1b. Figure 3: I suggest that the authors replace the red lines with a semi-transparent polygon. Figures 3 and 5: These figures can be integrated into one. Add lat-long, volcanoes,

rivers, and localities (for example. Santiago and Rancagua cities, San José de Maipo, etc) Figures 9, 11, 12 and 13: These figures will benefit a lot if, instead of presented the satellite image again, the authors add a geological map and locate the stereoplots in the borders of the map (not inside it).

---

## Referee Comment (RC2) · Anonymous Referee #2 · 25 Sep 2020

This paper deals with the structural characterisation of the Piuquencillo Fault System, a NW-striking structure intersecting the central-southern Andes. The authors provide an original structural dataset (orientation and kinematics of fault strands, orientation of hydrothermal veins, orientation of dikes and other intrusive contacts), coupled with U-Pb geochronology on zircons from two samples. The core of the manuscript is represented by stress inversion analysis on the structural dataset, to estimate paleostress tensor responsible for both fault architecture and fluid migration. The provided dataset has been integrated with geophysical one with the aim to discuss the importance of transversal structures as a pathway for endogenic fluids and as a potential seismic hazard in the Andean tectonic evolution. In my opinion, the scientific topic of this arti-

cle is of interest for a broad audience and the paper is suitable for publication on Solid Earth. One of the main strengths of the paper is the new structural dataset provided. On the other hand, the paper suffers from several limitations for what concerns both the text organisation and the data discussion/interpretation.

My major comments follow here, whereas I refer the authors to the attached file for my minor comments.

Text organisation 1) The Introduction should be broadened by emphasising the importance of a multidisciplinary approach in characterising the evolution (in time and space) of lithospheric-scale faults. Moreover, it should be clarified how it is possible to link structural information from exposed structures to deeply-seated tectonic lineaments. Some examples around the world should be mentioned for reference. Moreover, it should be emphasised the importance of constraining the time of tectonic evolution for the lithospheric-scale faults, to link a tectonic event to a hydrothermal/magmatic/volcanic process. 2) The Geological Background should be improved. In particular, I propose to better describe: (i) the subduction framework controlling the geodynamics of South America; (ii) the tectonic setting of South America (Western Cordillera, Eastern Cordillera, Coastal Range); (iii) the tectonic relationships between on-shore and off-shore occurrence of regional fault systems. This is recommended to better follow the tectonic framework illustrated within paragraph #5.3 and figures 16-17. 3) Results. I believe this paragraph should be improved by reorganising the text in three sub-paragraphs: a. #4.1 - Study areas (lines 113-116; lines 128-142) b. #4.2 - Structural analysis (lines 110-114; 144-165) c. #4.3 - U-Pb geochronology (lines 117-127; lines 532-538) Therefore, Figures 4 to 11 should be renumbered accordingly.

Data presentation/interpretation 1) I recommend providing structural constraints for the kinematics of the measured faults. The authors mentioned the faults are characterised by "low pitch angles, indicating predominantly strike-slip movements" (line 165). It is important to document what is reported in lines 203-204 (Syn-mineral displacement of the faults was mainly dextral for ENEtoNE-striking faults, sinistral for WNW-striking

faults, and sinistral-reverse for NNW-striking faults). Figure 4, alone, does not provide enough information. 2) Within the dynamic analysis, it should be clarified that the orientation of $\sigma1$ should bisect the acute angle between the two system faults (sinistral WNW-ESE-striking faults and dextral ENE-WSW-striking faults) that are considered conjugate (lines 201-202) by the authors. Therefore, the resulting $\sigma1$ should be more E-trending. 3) The relationships between the fault system and magmatic/hydrothermal products are not clear to me. Are the hydrothermal veins syn-kinematic to the conjugate fault systems? Do you have constraints about the ages of dikes? Is it possible to consider more than one generation of dikes (at least for those that are misoriented to the estimated stress regime)?

I hope that these comments and suggestions can improve the scientific quality of the present manuscript.

Sincerely

Please also note the supplement to this comment:
https://se.copernicus.org/preprints/se-2020-142/se-2020-142-RC2-supplement.pdf

———————————————————

[Figure]

**Supplement:**

[revised manuscript text omitted]

---

## Author Comment (AC1) · 2 Oct 2020

Dear Dr. Giambiagi, First of all, we would like to thank you very much for you valuable inputs to our manuscript. They will help us a lot to improve its quality. Regarding the main points highlighted in your letter: 1) We will include a broader tectonic context in the introduction, as suggested 2) We will add a discussion and additional references, to explain why we think this fault system is of lithospheric scale 3) Regarding the use of the kinematic and dynamic analyzes, we preferred to do both to be able to compare the results of these two different approaches and, from that, provide a stronger support to our interpretations. We will add a discussion about the possible perturbations/rotations

of the stress tensor in the vicinity of a major fault. We followed the suggestion of exploring possible differences in the area of Piuquencillo Alto between structural stations located at or close to the main NW-striking faults, and those located further away. However, the results do not show major differences, in both cases indicating a strike-slip regime with ENE-trending shortening and NNW stretching (see figure attached). Furthermore, no major differences are observed in the orientation of the kinematic axes or the calculated paleo-stress tensor between Piuquencillo Alto and the Maipo sector, which is located further away from the main branches of the PFS. The results obtained are also consistent with regional calculations of the stress tensor during the Miocene, presented in previous publications (Piquer et al., 2016). All of this is consistent with the fact that none of the cropping-out branches of the PFS is individually a major fault; we interpret that they represent the manifestation at the present-day surface of a major fault in the Andean basement, but the strain associated to each of the individual faults we mapped in the field is of small magnitude. Therefore, we conclude that there are no major perturbations of the regional stress tensor related to the individual fault traces of the PFS 4) We will include a more detailed discussion about the timing of dikes and veins, and why we can assign them a middle to late Miocene age. However, this can only be done based on cross-cutting relationships and correlations; there are no radiometric ages of the dikes emplaced along the PFS or of the vein system. We only have U-Pb ages of major plutons, and there are previously-published K-Ar and U-Pb ages of volcanic rocks. We attempted to obtain U-Pb zircon ages from the dike swarms, however no zircons were found in any of the collected samples. Regarding the veins, we also attempted to obtain an Ar/Ar age from syn-tectonic hydrothermal actinolite in Piuquencillo Alto, however, the calculated ages are not geologically possible; they are several My older than the U-Pb ages of the intrusive unit which hosts the veins 5) We will add a clearer explanation of why we think ENE-striking faults were more favorably oriented for dilation than NW-striking faults. As explained in the text, we think both sets of faults are part of large-scale, pre-existing fault systems (the PFS and the Yeso Valley faults). These fault systems acted broadly as conjugate faults under the prevailing

Miocene stress tensor, but they were not originated as such; because of this, they are not oriented at the ideal angle with respect to $\sigma 1$ expected in intact rocks, and one of the fault sets is more parallel to $\sigma 1$ than the other. We will add a new figure showing the strike of faults with a sinistral and dextral component, and we will also add a table with all the data from our structural stations, as a supplementary file 6) We agree that "stress reversal" is probably not appropriate wording; we do not intend to say that an extensional stress regime is established regionally during co-seismic periods. Our point is that, as is also mentioned in your letter, co-seismic dilation cause a drop in the normal stress in faults broadly perpendicular to $\sigma 1$, which, particularly under high fluid pressures, can lead to fault activation with normal kinematics (as shown recently by the activation of the Pichilemu fault after the Maule earthquake) We will also look carefully at all the minor comments mentioned in the letter. Kind Regards and thank you again for your careful review
* * *
[Figure]

[Figure]

Piuquencillo Alto
Away from main faults

N = 6

Piuquencillo Alto
Main faults

N = 13

**Fig. 1.**

---

## Author Comment (AC2) · 2 Oct 2020

Dear anonymous referee, First of all, we would like to thank you very much for you valuable inputs to our manuscript. They will help us a lot to improve its quality. Regarding the main points highlighted in your letter: - About the text organization: 1) and 2) We will enhance the Introduction and the Geological Background sections, including all the topics mentioned in your letter 3) We will follow your suggestion of presenting the U-Pb geochronology results as a separate sub-section, within the Results chapter - About data presentation and interpretation: 1) Regarding the first point (low pitch angle, mentioned in line 165), we will cite Fig. 10, which shows the pitch angle of the slickenlines

for all the measured fault planes. Regarding the preferred orientation of faults with different kinematics, we will add a new figure showing the strike of faults with a sinistral and dextral component, as suggested 2) Regarding the "conjugate" character of ENE- and NW-striking faults, we will add a clearer explanation of this topic. As explained in the text, we think both sets of faults are part of large-scale, pre-existing fault systems (the PFS and the Yeso Valley faults). These fault systems acted broadly as conjugate faults under the prevailing Miocene stress tensor, but they were not originated as such; because of this, they are not oriented at the ideal angle with respect to $\sigma1$ expected if they were newly-generated faults in intact rocks. In particular, the ENE-striking faults are more parallel to $\sigma1$ than the NW-striking ones. 3) Regarding the first question, yes, hydrothermal veins are syn-tectonic. We will specifically include a mention of this when describing the San Pedro de Nolasco veins, and will include in Fig. 4 field photographs of the main veins, to illustrate its syn-tectonic character. Regarding the age of dikes, we will include a more detailed discussion about the timing of their emplacement, and why we can assign them a middle to late Miocene age. However, this can only be done based on cross-cutting relationships and correlations; there are no radiometric ages of the dikes emplaced along the PFS. We only have U-Pb ages of major plutons, and there are previously-published K-Ar and U-Pb ages of volcanic rocks. We attempted to obtain U-Pb zircon ages from the dike swarms, however no zircons were found in any of the collected samples. Regarding the second question, yes, it is possible that there were several generations of dikes emplaced along the PFS, within the middle to late Miocene timeframe. We will also explicitly mention this in the text. We will also look carefully at all the minor comments mentioned in the supplement, and we will correct our manuscript accordingly. Kind Regards and thank you again for your careful review

---

## Author Response (AR1)

First of all, we would like to sincerely acknowledge the referees for their detailed reviews of our manuscript, which will certainly allow us to improve its quality. We have carefully considered all of their comments and suggestions and outlined our responses below, detailing how we have modified the text and figures accordingly.

**Referee 1: Laura Giambiagi**

**Comment:** *1) Tectonic setting: To give the manuscript a broader impact, I suggest adding a tectonic setting section, which can be integrated into the Geological-tectonic setting, with a synthesis of the Cenozoic extensional and compressional phases, and a description of the main structures of the area. Also, I would add a description of the Miocene intrusive complexes, such as the La Obra batholith.*

**Author´s response:** in the new version of the manuscript, the tectonic context is more extensively discussed in the Geological Background section, included an expanded synthesis of the main extensional and compressional phases, and the main fault systems active during these periods. The descriptions of the La Obra batholith and the Carlota intrusive complex were expanded at the Results section.

**Comment:** *2) Lithospheric-scale structures: Since there is no evidence that these fault systems involve the continental lithosphere, it is more convenient to name them as "continental-scale structural systems" (as they are called in the Discussion section). An alternative approach could be to use the available geophysical data and clearly propose the connection between the crustal structures and deeply-seated anisotropies. These geophysical data/analyses are just mentioned, but not properly presented as a discussion.*

**Author's response:** a new discussion and additional references were added to the text, to explain why we think this fault system is of lithospheric scale: *"Even though the whole lithosphere involvement of these continental-scale structures has not been demonstrated empirically in the Andes, several lines of reasoning provide independent arguments to support this concept. In one hand, Yáñez and Rivera (2019) proposed that the origin of these deep-seated structures could be related with master and transform faults associated with ancient rifting and/or suture zones related to collisional processes. In both likely scenarios, recent analogues show the presence of deep seated structures that involve the whole lithosphere (i.e. Kuna et al., 2019; Hua et al., 2019). On the other hand, several authors have demonstrated that continental-scale deformation zones, of some hundreds of kilometres length (comparable in scale to the PFS/MDZ), are controlled by the rheology of the mantle (i.e. Bird and Piper, 1980, England and McKenzie, 1984). Moreover, numerical models, in agreement with field observations, indicate that deformation decays laterally to 1/10 of the structure length for strike-slip dominated movements (England et al., 1985), thus developing a deformation zone of 10-30 km width, similar to the PFS/MDZ."*

**Comment:** *3) Kinematic vs dynamic analysis of fault-slip data: This must be more deeply discussed. Why do the authors perform these two kinds of analyses? What are the reduced stress tensors obtained from fault-slip data close to a major long-lived trans-crustal fault system telling us in terms of dynamics? Can they be interpreted as reflecting the stress of the crust during the movement of these structures? Since these structures have been previously generated during at least the Oligocene extension, or probably before (as the authors mention in the Discussion), these*

*tensors are not probably reflecting a stress field because, in this case, one of the basic assumptions during the inversion technique, the one concerning the absence of interference between faults, is not properly fulfilled. I suggest that the authors discriminate between stations close to the main faults, from stations located far away from the main faults. For example, data from the Piuquencillo Alto area could be separated into two (close and apart from the NW fault).*

**Author´s response:** regarding the use of the kinematic and dynamic analyzes, we preferred to do both to be able to compare the results of these two different approaches and, from that, provide a stronger support to our interpretations. A new discussion about the possible perturbations/rotations of the stress tensor in the vicinity of a major fault was added (*"When interpreting the results of stress tensor calculations from the inversion of fault slip data, a possibility which has to be considered is that stress tensor rotations might occur in the vicinity of major faults, although the expected patterns of stress rotation are still a matter of debate (Hardebeck and Michael, 2004; Famin et al., 2014). If these stress tensor rotations occurred at the PFS, then part of our calculations might not represent a regional stress field, but a local stress tensor acting only in and around the fault traces. The dynamic analysis by sector, however, suggests that this is not the case. The Piuquencillo Alto and Piuquencillo-Claro sectors cover areas around the main traces of the PFS, while the Maipo sector is located further away from it (Fig. 13). In these three sectors, the direction of maximum horizontal compression is similar (E-W to slightly ENE), without any evidence of rotations occurring around the traces of the PFS. The results obtained are also consistent with regional calculations of the Miocene – early Pliocene stress field in central Chile (Piquer et al., 2016). All of this is consistent with the fact that none of the cropping-out branches of the PFS is individually a major fault; the strain associated to each of them is of small magnitude, so no major perturbations of the stress tensor are expected around them"*). We followed the suggestion of exploring possible differences in the area of Piuquencillo Alto between structural stations located at or close to the main NW-striking faults, and those located further away. However, the results do not show major differences, in both cases indicating a strike-slip regime with ENE-trending shortening and NNW stretching. Furthermore, no major differences are observed in the orientation of the kinematic axes or the calculated paleo-stress tensor between the Piuquencillo Alto, Piuquencillo-Claro and Maipo sectors, despite the fact that the latter is located further away from the main branches of the PFS. The results obtained are also consistent with regional calculations of the stress tensor during the Miocene – early Pliocene, presented in previous publications (Piquer et al., 2016). All of this is consistent with the fact that none of the cropping-out branches of the PFS is individually a major fault; we interpret that they represent the manifestation at the present-day surface of a major fault in the Andean basement, but the strain associated to each of the individual fault observed in the field is of small magnitude. Therefore, we conclude that there are no major perturbations of the regional stress tensor related to the individual fault traces of the PFS.

**Comment:** *4) Age of dikes and intrusives: You should clarify the relationship between the emplacement of dikes and intrusives, and movement along the study fault system. Are there any time constraints on this relationship? For example, what is the age of the Ag-Pb-Zn-Cu vein system in the San Pedro de Nolasco sector? In this respect, move paragraph 182-186 from the Discussion section to the Results section, and add any other information about the timing of emplacement of these intrusives and the timing of slips of the studied faults.*

**Author's response:** a more detailed discussion was included about the timing of dikes and veins, and why we can assign them a middle to late Miocene age. However, this can only be done based on cross-cutting relationships and correlations; there are no radiometric ages of the dikes emplaced along the PFS or of the vein system. We only have U-Pb ages of major plutons, and there are previously-published K-Ar and U-Pb ages of volcanic rocks. We attempted to obtain U-Pb zircon ages from the dike swarms, however no zircons were found in any of the collected samples. Regarding the veins, we also attempted to obtain an $^{40}Ar/^{39}Ar$ age from syn-tectonic hydrothermal actinolite in Piuquencillo Alto, however, the calculated ages are not geologically possible; they are several My older than the U-Pb ages of the intrusive unit which hosts the veins. Regarding the paragraph in lines 182-186, this is now included in the Discussion section, following the suggestion of the Topical Editor regarding moving the paleo-stress tensor calculations entirely to the Discussion chapter.

**Comment:** *5) The fluid pathways: This idea must be broadened. Discuss here how sigma 1 is obtained. Are WNW and ENE conjugated strike-slip faults, or not? For each measurement station, add the main strike of dextral and sinistral faults, as well as the orientation of local stress tensor axes, and then compare them. As I pointed out above, it would be nice to have had previously discrimination between tensors obtained close to or far apart from the main faults. It is not clear to me why the ENE striking faults are optimally-oriented for the migration of fluids, while the WNW striking faults are not. If both sets are conjugate strike-slip faults, as stated in lines 201-202, why is one set more likely to dilate and the other one more likely to slip? In my opinion, more information about the strike of the local sigma 1 is needed to evaluate this proposal. Add a table containing: location of each station, number of measured faults, results of fault-slip inversion (orientation of principal stress axes and stress ratio), and lapse of time for each calculated reduced stress tensor.*

**Author's response:** regarding the first point, as mentioned before, we added a discussion regarding the possible variations in the stress tensor between structural stations located close to or far apart from the main branches of the PFS. A new figure was added showing the strike of faults with a sinistral and dextral component (Fig. 16 in the new version of the manuscript), and a table with all the data from our structural stations is now included as Supplementary material. We added a clearer explanation of why we think ENE-striking faults were more favorably oriented for dilation than NW-striking faults (*"This suggests that the PFS and the ENE-striking faults acted broadly as conjugate faults under the prevailing middle Miocene – early Pliocene stress tensor. However, they are not oriented at the ideal angle with respect to σ$_1$ expected in intact rocks, with the ENE-striking faults being more parallel to σ$_1$ than the fault planes of the PFS. The reason for this might be that both sets of faults are part of large-scale, pre-existing fault systems (the PFS and, for the ENE-striking faults, the Yeso Valley Fault System), reactivated during the Mio-Pliocene, but not originated as conjugate structures. As the ENE-striking faults are more parallel to the predominant orientation of σ$_1$ (E-W to ENE-trending), they were the most favourably oriented for opening, which explains why ENE-striking veins are as common as those striking WNW, while ENE-striking fault planes are much less frequent"*). As explained in the text, we think both sets of faults are part of large-scale, pre-existing fault systems (the PFS and the Yeso Valley faults). These fault systems acted broadly as conjugate faults under the prevailing Mio-Pliocene stress tensor, but they were not originated as such; because of this, they are not oriented at the ideal angle with respect to σ$_1$ expected in intact rocks, and one of the fault sets is more parallel to σ$_1$ than the other.

**Comment:** *6) Transient stress reversals: This statement is very questionable. It is difficult to explain a change from compression to extension as stress reversals during the coseismic stress release. Positive changes in the Coulomb failure stress (see Harris et al. 1998; Stein, 1999; King and Cocco, 2000; Freed 2005) bring receiver faults in the continental crust closer to failure, but the magnitudes involved in these stress changes are small (<1 MPa). The process is more related to the unclampling of the receiver faults, due to a drop in the normal stress that prevents the slip during the interseismic period. See Spagnotto et al (2015), who explore how changes in the Coulomb conditions associated with the Maule earthquake triggered upper-plate earthquakes. They argue that the reactivation of faults after a mega-earthquake at the subduction interphase derives from unclamping processes associated with co-seismic dilatation deformation inferred from GPS observations.*

**Author's response:** we agree that "stress reversal" is probably not appropriate wording; we changed it to "stress relaxation, leading to transient local extension" in the new version of the manuscript. We added a reference to the very interesting work of Spagnotto et al. (2015). We do not intended to say that an extensional stress regime is established regionally during co-seismic periods. Our point is that, as is also mentioned in the referee's comment, co-seismic dilation cause a drop in the normal stress in faults broadly perpendicular to $\sigma_1$, which, particularly under high fluid pressures, can lead to fault activation with normal kinematics (as shown recently by the activation of the Pichilemu fault after the Maule earthquake).

All the corrections included as "minor comments" in the letter were incorporated to the manuscript, with the following exceptions:

**Line 75:** *Here I wonder how are these sheeted dikes related to mineralization in the El Teniente porphyry?*

**Author's response:** as explained before, these dikes aren't related to mineralization at El Teniente; they are older than the late Miocene hydrothermal alteration affecting the Carlota Intrusive Complex.

**Line 103:** *Reduced paleo-stress tensors: Briefly describe here which is the methodology to separate heterogeneous data sets into homogeneous ones, since this is a very critical step to obtained robust tensors.*

**Author's response:** as explained in the text, for paleo-stress tensor calculations we used the Multiple Inverse Method, which allows to work directly with heterogeneous data sets, and to identify the different stress tensors under which the considered fault planes were active.

**Lines 277-278:** *The focal mechanism solution for the Las Melosas earthquake is quite particular, with NNW-oriented P axis, not compatible with the WNW-striking sinistral faults of the PFS.*

**Author's response:** one of the focal mechanism solutions for the Las Melosas earthquake is a subvertical fault striking N74°W, highly coincident with the PFS. However, it is correct that the sense of movement of this particular fault reactivation would be dextral strike-slip, not sinistral as was the predominant sense of movement during the Mio-Pliocene. We added a mention to this point in the new version of the text.

Comments about the figures:

**Figure 1:** *add sense of displacement of normal and strike-slip faults.*

**Author's response:** the faults shown in this figure, in particular de fundamental basement fault, can have multiple reactivations trough time with different kinematics, determined by variations on the orientation and relative magnitudes of the principal stresses. Because of this, it is not possible to assign a specific sense of movement to the faults shown in the diagram.

**Figure 2:** *Improve the location map (a) to show the main tectonic characteristics of the study area. Add lat- long in the map from Figure 1b.*

**Author's response:** done, as suggested by the reviewer.

**Figure 3:** *I suggest that the authors replace the red lines with a semi-transparent polygon.*

**Author´s response:** done, as suggested by the reviewer.

**Figures 3 and 5:** *These figures can be integrated into one. Add lat-long, volcanoes, rivers, and localities (for example. Santiago and Rancagua cities, San José de Maipo, etc)*

**Author's response:** we decided to leave them as separate figures, but showing the Landsat satellite image in Figure 3 and the geological map in Figure 5. In Figure 3, we labelled the cities of Santiago and Rancagua, the Maipo river valley, and the Maipo volcano, as suggested.

**Figures 9, 11, 12 and 13:** *These figures will benefit a lot if, instead of presented the satellite image again, the authors add a geological map and locate the stereoplots in the borders of the map (not inside it).*

**Author's response:** we replaced the satellite image by the geological maps in all these figures, as suggested by the reviewer.

**Referee 2: anonymous**

***About the text organization:***

**Comment**: *1) The Introduction should be broadened by emphasising the importance of a multidisciplinary approach in characterising the evolution (in time and space) of lithospheric-scale faults. Moreover, it should be clarified how it is possible to link structural information from exposed structures to deeply-seated tectonic lineaments. Some examples around the world should be mentioned for reference. Moreover, it should be emphasised the importance of constraining the time of tectonic evolution for the lithospheric-scale faults, to link a tectonic event to a hydrothermal/magmatic/volcanic process.*

**Author's response:** the introduction was enhanced, including all the topics mentioned in the referee's comment.

**Comment:** *2) The Geological Background should be improved. In particular, I propose to better describe: (i) the subduction framework controlling the geodynamics of South America; (ii) the tectonic setting of South America (Western Cordillera, Eastern Cordillera, Coastal Range); (iii) the*

*tectonic relationships between on-shore and off-shore occurrence of regional fault systems. This is recommended to better follow the tectonic framework illustrated within paragraph #5.3 and figures 16-17.*

**Author's response:** in the new version of the text, these points were all included in the Introduction and the Geological Background.

**Comment:** *3) Results. I believe this paragraph should be improved by reorganising the text in three sub-paragraphs: a. #4.1 - Study areas (lines 113-116; lines 128-142) b. #4.2 – Structural analysis (lines 110-114; 144-165) c. #4.3 - U-Pb geochronology (lines 117-127; lines 532-538) Therefore, Figures 4 to 11 should be renumbered accordingly.*

**Author's response:** following the referee's suggestion, the U-Pb geochronology results are now presented as a separate sub-section, after the description of the different sectors into which the study area was subdivided, within the Results chapter.

***About data presentation and interpretation:***

**Comment:** *1) I recommend providing structural constraints for the kinematics of the measured faults. The authors mentioned the faults are characterized by "low pitch angles, indicating predominantly strike-slip movements" (line 165). It is important to document what is reported in lines 203-204 (Syn-mineral displacement of the faults was mainly dextral for ENE to NE-striking faults, sinistral for WNW-striking faults, and sinistral-reverse for NNW-striking faults). Figure 4, alone, does not provide enough information.*

**Author's response:** regarding the first point (low pitch angle), we added a reference to Fig. 10, which shows the pitch angle of the slickenlines for all the measured fault planes. Regarding the preferred orientation of faults with different kinematics, we added a new figure (Fig. 16 in the new version of the manuscript) showing the strike of faults with a sinistral and dextral component, as suggested.

**Comment:** *2) Within the dynamic analysis, it should be clarified that the orientation of $\sigma_1$ should bisect the acute angle between the two system faults (sinistral WNW-ESE-striking faults and dextral ENE-WSW-striking faults) that are considered conjugate (lines 201-202) by the authors. Therefore, the resulting $\sigma_1$ should be more E-trending.*

**Author's response:** please see our response to Comment 5) of Referee 1, which refers to this same tropic.

**Comment:** *3) The relationships between the fault system and magmatic/hydrothermal products are not clear to me. Are the hydrothermal veins syn-kinematic to the conjugate fault systems? Do you have constraints about the ages of dikes? Is it possible to consider more than one generation of dikes (at least for those that are misoriented to the estimated stress regime)?*

**Author's response:** regarding the first question, yes, hydrothermal veins are syn-tectonic. In the new version of the manuscript, we specifically mention this when describing the San Pedro de Nolasco veins, and we also included in Fig. 4 field photographs of the main veins, to illustrate its syn-tectonic character. Regarding the age of dikes, as mentioned in a similar comment by Referee 1 (comment 4), we included a more detailed discussion about the timing of their emplacement,

and why we can assign them a middle to late Miocene age. Regarding the second question, yes, it is possible that there were several generations of dikes emplaced along the PFS, within the middle to late Miocene timeframe. We now explicitly mention this in the text.

Regarding the minor comments mentioned in the supplement, all of the corrections and suggestions were incorporated to the new version of the manuscript, with the following exceptions:

**Line 94:** *I would like to remove Figure 4 from here, as it is part of the results from structural work.*

**Author's response:** we would prefer to leave the reference to Figure 4 in the Methodology section, as it illustrate to the reader the different types of kinematic criteria which were used to establish the sense of movement of faults.

**Lines 182 to 186:** *this part should move in the Results paragraph.*

**Author's response:** following the suggestion of the Topical Editor, all the section regarding paleo-stress tensor calculations was moved to the Discussion chapter.

Comments about the figures:

**Figure 3:** *I would put here a geological map as the background. The Landsat image is not useful to the aim of the paper. Moreover, the legibility of the fault pattern (black lines) is hindered by the coloured image.*

**Author's response:** we decided to leave the satellite image in this particular figure, and we added labels to show geographical features useful for the readers (main cities, valleys, the Maipo volcano), as suggested by Referee 1. However, we replaced the satellite image by the geological map in all the subsequent figures.

**Figure 4:** *these pictures need for the orientation. How was the sense of movement estimate for pictures (a) and (b)?*

**Author's response:** the picture orientation was added to the figure, as suggested by the Referee. The sense of movement was estimated from the geometry of steps in syn-tectonic epidote and actinolite, and from its crystallization in strain fringes. This explanation was added to the figure caption.

**Figure 5:** *same comment as before. I would replace the Landsat image with a geological map. The same for next figures*

**Author's response:** done as suggested by the reviewer.

**Figure 7:** *please, provide picture orientation and add more labels/lines to improve the legibility.*

**Author's response:** done as suggested by the reviewer.

[revised manuscript text omitted]

---

## Referee Report (RR1)

| (1) | |  |  |  |  |  |  |  | |
| (1) | |  |  |  |  |  |  |  | |
| (1) | |  |  |  |  |  |  |  | |
| (1) | |  |  |  |  |  |  |  | |

700

[referee-annotated manuscript omitted]

---

## Author Response (AR2)

We would like to sincerely acknowledge again the referees for their careful reviews and encouraging words about our manuscript.

Apart from the referee's suggestions, we have also added larger labels to the kinematic axes in Figures 10, 12 and 14, and we included scale bars in kilometers on Figures 16 and 17.

**Referee 1: Dr. Laura Giambiagi**

We have incorporated to the text and figures all the minor suggestions made by the referee.

**Referee 2: Dr. Gianluca Vignaroli**

**Comment**: *In addition to my minor suggestions, I would like to reduce the number of figures. I suggest moving Figures 9, 11 and 12 into the supplementary material, as they show data projection on the same geological map of Figure 5.*

**Author's response:** we understand the referee's concerns, regarding the relatively large number of figures included in our manuscript. However, we are convinced that it is important to keep Figures 9, 11 and 12 as part of the main text, as they illustrate the spatial variations of the preferred orientations of fault planes and of the kinematic axes in different sectors of the Piuquencillo Fault System. However, after reviewing the rest of the figures, we have decided to merge into one the previous Figures 14 and 15 (Fig. 14 in the new version of the manuscript). This also solves one of the minor comments made by the referee, regarding an excess of blank background in the same two figures.

All the minor suggestions made by the reviewer directly in the pdf file have been incorporated, with the following exception:

**Figure 7:** *could you show some traces of folded layers at the bottom of the unconformity?*

**Author's response:** sadly, bedding is not clearly visible from this angle in the area right below the unconformity. The only area with clear bedding at the Farellones Formation is at the far right of the picture, several meters below the unconformity.

[revised manuscript text omitted]